# Boosting Multiple Views for pretrained-based Continual Learning

**Quyen Tran**[1†], **Lam Tran**[1†], **Khanh Doan**[1], **Toan Tran**[1], **Dinh Phung**[3], **Khoat Than**[2*], **Trung Le**[3*]

[1] Qualcomm AI Research**    [2] Hanoi University of Science and Technology    [3] Monash University

## Abstract

Recent research has shown that Random Projection (RP) can effectively improve the performance of pre-trained models in Continual learning (CL). The authors hypothesized that using RP to map features onto a higher-dimensional space can make them more linearly separable. In this work, we theoretically analyze the role of RP and present its benefits for improving the model's generalization ability in each task and facilitating CL overall. Additionally, we take this result to the next level by proposing a *Multi-View Random Projection scheme* for a stronger ensemble classifier. In particular, we train a set of linear experts, among which diversity is encouraged based on the principle of AdaBoost, which was initially very challenging to apply to CL. Moreover, we employ a task-based adaptive backbone with distinct prompts dedicated to each task for better representation learning. To properly select these task-specific components and mitigate potential feature shifts caused by misprediction, we introduce a simple yet effective technique called the self-improvement process. Experimentally, our method consistently outperforms state-of-the-art baselines across a wide range of datasets.

## 1 Introduction

Continual Learning (CL) is a field of Machine Learning that focuses on enabling deep neural networks to learn continually from a sequence of tasks with different data distributions. Sequential training on such data often leads to *catastrophic forgetting* (French, 1999) of old knowledge, where model parameters are overwritten by new tasks's learning. To overcome this problem, recent CL methods proposed leveraging the generalizability of pre-trained models (Han et al., 2021; Jia et al., 2022) as frozen backbones to continually solve the series of emerging tasks overtime (Janson et al., 2022; Wang et al., 2022c; Zhou et al., 2023; Smith et al., 2023; Li et al., 2024).

Among these pretrained-based CL methods, one promising work is RanPAC (McDonnell et al., 2023), which is based on Random projection (RP) to facilitate the classification ability of models. In particular, this method proposes using a random matrix to project the features obtained from a pre-trained backbone onto a new space with significantly higher dimensions (e.g., from $d = 768$ to $d' = 10,000$). The motivation for utilizing RP is based on their empirically supported but not explicitly theoretically proven hypothesis, that features are more linearly separable and easier to classify in higher-dimensional space. In our work, we theoretically explain the roles of RP followed by a nonlinear activation function, showing that the margin of a training data point in the higher-dimensional space is almost surely larger by the rate of $O(\sqrt{d'})$. This means that RP increases the (nonlinear) separability between classes and tasks. We also analyze RP's implications regarding generalization ability for each specific task and facilitating CL overall.

In addition, to take advantage of RP for enhancing separability, a straightforward solution is to set the dimension of the projected space (i.e., $d'$) to be significantly large. However, this naive idea would be infeasible as training the classifier involves computing the inverse of a Gram matrix of size $d' \times d'$. To overcome this obstacle while leveraging the multiple-view learning paradigm for

---

[†,*]Equally contributed.
**Qualcomm Vietnam Company Limited

a stronger classifier, we instead consider handling $K$ projected spaces with smaller manageable dimensions and employing $K$ corresponding projection matrices. We refer to each space as an *atomic view* (i.e., $d'_a$). During training, the classifiers on these views will be encouraged to diversify through exploiting the principle of Adaboost (Freund & Schapire, 1997) in CL for the first time. We also theoretically argued that computations on a huge view, which is composed of $k \leq K$ atomic views, could be conducted through similar operators on these smaller views by some relaxation. Therefore, given the responses from $K$ atomic views, we can generate up to $2^K - 1$ huge-view responses and implement a voting strategy to obtain the final prediction, harnessing the potential of the boosting algorithm in CL.

Furthermore, we argue that the first task adaptation strategy (Panos et al., 2023) in RanPAC, while bridging the gap between pre-trained data and CL tasks, may cause overlapping representations from later tasks, potentially limiting model performance as subsequent tasks may differ from the first. To effectively adapt incoming tasks, we follow prompt-based CL methods (Wang et al., 2022c; Smith et al., 2023), assigning each task to different prompts and utilizing the prompted model to obtain features for RP. However, we observe that occasionally, incorrect prompt selection can lead to feature shifts between the training and testing phases, resulting in poor performance. Therefore, to mitigate the risk of selecting the wrong prompt, we propose a self-improvement process for appropriately selecting prompts for each task sample.

Finally, we name our method as "***Boost**ing Multiple Views for pretrained-based **C**ontinual **L**earning (BoostCL)*", and summarize our main contributions as follows:

- We theoretically analyze the benefits of random projection (RP) mapping representations onto a higher-dimensional space, which is explicitly unsolved in the previous work. We show that RP increases the margin of each training instance, making features more separable. Besides, RP can increase a model's generalization ability for each specific task and facilitate CL overall.

- Based on our theoretical results, to handle a vastly high-dimensional space, we propose a novel *Multi-View random projection scheme* for training a strong ensemble classifier. This scheme involves leveraging an adaptive AdaBoost strategy, which initially poses a significant challenge when applied directly to CL.

- Additionally, together with integrating task-specific prompts into the CP-based backbone for better adaptation, we propose a self-improvement process, a simple but effective strategy to help select prompts more accurately when inference.

- We empirically evaluate the effectiveness of our method against current state-of-the-art pre-trained-based baselines across various benchmarks. Specifically, denoting *BoostCL-m(d′;K)* as our approach with $K$ views and $d'$ dimensions for each view, we consider the following variants: *BoostCL-m* ($d' = 768; K = 13$), *BoostCL-m* ($d' = 10,000; K = 15$), and *BoostCL-m* (without random projection;$K = 15$) for comparison with RandAC and other baselines. Our variant *BoostCL-m* ($d' = 10,000; K = 15$) outperforms all baselines by a significant margin. Interestingly, *BoostCL-m* ($d' = 768; K = 13$), which uses the same random-projection dimension as RanPAC, also surpasses the baseline by a remarkable margin, showcasing the effectiveness of our multi-view learning strategy. Notably, the variant *BoostCL-m* (without random projection;$K = 13$), applying multi-view without random projection, also significantly outperforms RanPAC and other baselines, reinforcing the value of our multi-view learning approach.

## 2 RELATED WORK

**Class Prototype-based approach** leverages pre-trained models and introduces prototype-based strategies for continual learning (CL). In particular, the nearest class mean (NCM) classifier applied to prototypes from frozen pre-trained models is a strong baseline (Janson et al., 2022). Later, ADaM (Zhou et al., 2023) refines this by adapting the pre-trained model to the first task with techniques like prompt-tuning (Jia et al., 2022) or adaptors (Chen et al., 2022). Building on this, recent work (McDonnell et al., 2023; Goswami et al., 2023; Zhuang et al., 2023; ZHUANG et al., 2022) further enhance NCM by using second-order statistics, resulting in advanced classifiers with closed-form solution.

**Prompt-based approach.** Wang et al. (2022d;c); Smith et al. (2023) typically assigns a set of prompts to each task, enhancing the adaptability of the backbone to downstream tasks and enabling the ability to distinguish classes within each task. However, the absence of explicit constraints can lead to feature overlapping between classes from new and previous tasks. Therefore, recent methods employ some types of contrastive loss (Wang et al., 2023; Li et al., 2023) or utilize Vision Language models (Wang et al., 2022a; Nicolas et al., 2024) to better separate features from tasks.

Although the methods in these two approaches achieve impressive performance, they only consider classifiers with a single view, resulting in a limited performance. To create a stronger classifier, we propose a novel Multi-view Random Projection scheme with an ensemble classifier, including many diverse views driven by the principle of Adaboost (Freund & Schapire, 1997), where responses from sub-classifiers of different views are then combined and voted on to produce the final prediction.

## 3 BACKGROUND

### 3.1 PROBLEM FORMULATION

We consider Continual Learning setting, where a model learns from a sequence of $T$ classification tasks without revisiting old task data during training or task ID access during inference. Each task $t \in \{1, ..., T\}$ has dataset $D_t$ containing $n_t$ i.i.d. samples $(\boldsymbol{x}_i^t, y_i^t)_{i=1}^{n_t}$ of $\mathcal{Y}_t$ classes.

In this work, we design our model with the following components: a pre-trained ViT backbone $f_\Phi(\cdot)$ parameterized by $\Phi$; a set of frozen random projection matrices $\{\boldsymbol{U}^i\}_{i=1}^K$, and $\boldsymbol{U}' \in \mathbb{R}^{d \times d'} (d' \gg d)$, where $K$ is the total number of atomic views, $d$ and $d'$ are the dimensions of original and projected latent space, respectively; and two classification heads with weight matrices $\boldsymbol{W}, \boldsymbol{W}'$ for class predictor and prompt predictor, respectively. Similarly to other prompt-based works, we incorporate a set of prompts $\mathbf{P}$ into $f_\Phi$ and denote the backbone after incorporating the prompts as $f_{\Phi, \boldsymbol{P}}$.

### 3.2 THE CLOSE-FORM SOLUTION FOR THE CL CLASSIFIER OF RANPAC

RanPAC (McDonnell et al., 2023) is a CP-based method that proposes using a random projection matrix $\boldsymbol{U} \in \mathbb{R}^{d \times d'}$ to project feature vector $\boldsymbol{z} = f_\Phi(\boldsymbol{x}) \in \mathbb{R}^{1 \times d}$ of each sample $\boldsymbol{x}$ onto a significantly higher dimensional space and then apply a nonlinear activation function $a$ to obtain representation $\tilde{\boldsymbol{z}} = a(\boldsymbol{z}\boldsymbol{U}) \in \mathbb{R}^{1 \times d'}$. Let $\boldsymbol{X} = \{\boldsymbol{x}_i\}_{i=1}^{N_t}$ be the set of samples up to task $t$ where $N_t = \sum_{i=1}^t n_i$. The corresponding projected representations and labels are $\tilde{\boldsymbol{Z}} \in \mathbb{R}^{N_t \times d'}$ and $\boldsymbol{Y} \in \mathbb{R}^{N_t \times K_t}$, where $K_t$ is the total number of classes up to task $t$. This method aims to learn a linear classifier $\boldsymbol{W} \in \mathbb{R}^{d' \times K_t}$ that can obtain knowledge from all tasks so far without forgetting by solving the following problem:

$$\min_{\boldsymbol{W}} \|\tilde{\boldsymbol{Z}}\boldsymbol{W} - \boldsymbol{Y}\|_2^2 + \frac{\lambda}{2}\|\boldsymbol{W}\|_2^2. \tag{1}$$

The closed-form solution of (1) is $\boldsymbol{W} = (\tilde{\boldsymbol{Z}}^T \tilde{\boldsymbol{Z}} + \lambda \mathbf{I})^{-1} \tilde{\boldsymbol{Z}}^T \boldsymbol{Y}$, which can be updated incrementally across the arriving tasks. Specifically, let $\boldsymbol{G}_t = \tilde{\boldsymbol{Z}}^T \tilde{\boldsymbol{Z}} \in \mathbb{R}^{d' \times d'}$ and $\boldsymbol{C}_t = \tilde{\boldsymbol{Z}}^T \boldsymbol{Y} \in \mathbb{R}^{d' \times K_t}$ be the Gram matrix and class prototype matrix computed at the end of task $t$, respectively. We have:

- $\boldsymbol{G}_t = \boldsymbol{G}_{t-1} + \tilde{\boldsymbol{Z}}_t^T \tilde{\boldsymbol{Z}}_t$ where $\boldsymbol{G}_1 = \tilde{\boldsymbol{Z}}_1^T \tilde{\boldsymbol{Z}}_1$, and $\tilde{\boldsymbol{Z}}_i$ is projected representations from task $i$.
- $\boldsymbol{C}_t = \boldsymbol{C}_{t-1} + \tilde{\boldsymbol{Z}}_t^T \boldsymbol{Y}$ where $\boldsymbol{C}_1 = \tilde{\boldsymbol{Z}}_1^T \boldsymbol{Y}_1$, and $\boldsymbol{Y}_i$ is one-hot vector labels from task $i$.

After task $T$, to classify $\boldsymbol{x}_{\text{test}}$, we simply perform: $y_{\text{pred}} = \text{argmax}_y \tilde{\boldsymbol{z}}_{\text{test}}^T (\boldsymbol{G}_T + \lambda \mathbf{I})^{-1} \boldsymbol{c}_y$, where $\boldsymbol{c}_y$ is the $y^{\text{th}}$ column of $\boldsymbol{C}_T$.

### 3.3 THE BOOSTING PRINCIPLE OF ADABOOST/SAMME

AdaBoost (Freund & Schapire, 1997) and SAMME (Hastie et al., 2009) are ensemble learning methods that combine multiple weak classifiers $\{g^k(\cdot)\}_{k=1}^K$ to create a strong classifier $G(\cdot)$ by sequential training on weighted training data $D = \{(\boldsymbol{x}_i, y_i)\}_{i=1}^N$. These methods focus on misclassified examples in each iteration $k$ to improve the model performance iteratively. Directly applying AdaBoost/SAMME to CL poses a significant challenge because the data for all tasks is not always available, and the computational cost for a sufficiently large number of weak learners is infeasible. Please refer to Appendix B for more details.

## 4 FRAMEWORK

In this part, we first analyze the theoretical role of RP and its implications (Section 4.1). Then, we introduce our proposed *Multi-view Random Projection strategy* for our classifier (Section 4.2). Finally, we explain how we adapt the CP-based backbone to new tasks and select task-specific components to improve accuracy (Section 4.3).

### 4.1 RANDOM PROJECTION ONTO HIGHER-DIMENSIONAL SPACE CAN IMPROVE INSTANCE MARGIN, GENERALIZATION, AND CL OVERALL

In RanPAC (McDonnell et al., 2023), the authors *hypothesized* that performing a random projection (RP) onto a higher-dimensional space can help the transformed features become more linearly separable. Nonetheless, they did not provide explicit proof, leaving the motivation and the hypothesis unsolved. In this section, we theoretically analyze the role of RP and show its vital implications.

**Improving margin.** According to Sokolić et al. (2017), given a classifier $g$, the margin of an instance $s = (\boldsymbol{z}, y)$ can be defined as $\gamma(s, g, \mathbb{R}^d) = \sup \{\nu : \|\boldsymbol{z} - \boldsymbol{z}'\| \leq \nu \Rightarrow g(\boldsymbol{z}') = y, \forall \boldsymbol{z}' \in \mathbb{R}^d\}$. When taking an RP with a transform matrix $\boldsymbol{U} \in \mathbb{R}^{d \times d'}$, each instance $s \in \mathbb{R}^d \times \mathcal{Y}$ will have a projection $s^u = (\boldsymbol{z}^T \mathbf{U}, y) \in \mathbb{R}^{d'} \times \mathcal{Y}$, where $\mathcal{Y}$ is the label space.

Given a training set $S$, we define the margin of $g$ over the training set $S$ as $\gamma(S, g, \mathbb{R}^d) = \min_{s \in S} \gamma(s, g, \mathbb{R}^d)$. The training set $S$ over $\mathbb{R}^d$ induces the training set $S^u$ over $\mathbb{R}^{d'}$ through the transform matrix $U$. Similarly, we define the margin over $S^u$. According to Sokolić et al. (2017) and our Corollary A.3 in Appendix A.1, a classifier with higher margins over a training set $S$ generalizes better to a general distribution from which instances in $S$ are sampled.

To see the benefits of RP, we propose the concept of *Bayes margin* to measure the highest margin of the classifiers in a hypothesis family.

**Definition 4.1** (Bayes margin). Let $\mathcal{H}$ be the set of measurable functions that map from $\mathbb{R}^d$ to $\mathcal{Y}$. The *Bayes margin* over the training set $S$ is defined as $\gamma_{\text{in}}(S, \mathcal{H}) = \sup_{g \in \mathcal{H}} \gamma(S, g, \mathbb{R}^d)$.

Note that $\mathcal{H}$ is large enough to cover all hypothesis spaces often used in machine learning. Using a smaller space may not correctly reflect the role of RP. Similarly, let $\mathcal{H}'$ be the set of measurable functions that map from $\mathbb{R}^{d'}$ to $\mathcal{Y}$. The Bayes margin of $S^u$ is: $\gamma_{\text{in}}(S^u, \mathcal{H}') = \sup_{h \in \mathcal{H}'} \gamma(S^u, h, \mathbb{R}^{d'})$. We have the following result about the role of RP (see Appendix A for proof and more discussions).

**Theorem 4.2.** *Let $\boldsymbol{U}$ be a random matrix of size $d' \times d$, with $d' \gg d$, whose elements are independent copies of a Gaussian random variable with unit variance and 0 mean. Consider $s = (\boldsymbol{z}, y)$ and its projection $s^u = (\sigma(\boldsymbol{z}^T \boldsymbol{U}), y)$ where the element-wise nonlinearity $\sigma$ is invertible and expansive.[1] For every $\epsilon \in (0, 1)$, with probability at least $1 - (C.\epsilon)^{d'-d+1} - e^{-c.d'}$ where $C$ and $c$ are some positive constants, we have:* $\gamma_{in}(S^u, \mathcal{H}') \geq \gamma_{in}(S, \mathcal{H})(\sqrt{d'} - \sqrt{d-1})\epsilon.$

According to this theorem, the Bayes margin of a sample in the new space can increase at a rate of $O(\sqrt{d'})$, *thus potentially enhancing generalization ability*. This property of RP is intriguing and has never been known before. As discussed next, this property should lead to significant consequences for each task of interest and task sequence.

**Implications to generalization ability and CL.** Denote $\mathcal{Z}_c^{class}$ and $\mathcal{Z}_t^{task}$ as the sample domains whose members belong to class $c$, and task $t$ respectively. The *Bayes margin of each class $c$* is defined as $\gamma_{\text{class},c} = \min\{\gamma_{\text{in}}(S, \mathcal{H}) : S \ni s = (\boldsymbol{z}, c), \boldsymbol{z} \in \mathcal{Z}_c^{class}\}$. The *Bayes margin of a task $t$* is defined as $\gamma_{\text{task},t} = \min\{\gamma_{\text{in}}(S, \mathcal{H}) : S \ni s = (\boldsymbol{z}, y), \boldsymbol{z} \in \mathcal{Z}_t^{task}\}$. The following implications readily follow:

- *RP can increase generalization ability for a task.* As shown in Theorem 4.2, the margin of every instance increases in the new space, resulting in $\gamma_{\text{class},c}$ and $\gamma_{\text{task},t}$ increase at a rate of $O(\sqrt{d'})$. Thus, classes in the new space become more separable. Moreover, there

---

[1]A function $\sigma$ is expansive if it satisfies $|\sigma(x) - \sigma(y)| \geq |x - y|$ for all $x, y$ in their domain.
Note: ReLU can work well probably because it approximates functions that satisfy the two conditions above. For example, $a(x) = x + \xi \cdot \log(1 + \exp(x))$ where $\xi \in \mathbb{R}$, which is used in our experiments (see Appendix G.4).

is a well-known connection between the margin and the generalization ability of a model (Bartlett et al., 2017; Sokolić et al., 2017). Consequently, the generalization error for each task $t$ will be smaller. This means the trained model can generalize better on unseen data. More discussions about this aspect can be found in Appendix A.

- *RP can increase task separability and CL overall.* As discussed above, the margin $\gamma_{\text{task},t}$ for each task $t$ increases, indicating rapid task separability in $\mathbb{R}^{d'}$. This improvement in task separability is crucial in CL, aiding task detection processes. Coupled with enhanced generalization ability for each task, this provides a theoretical basis for why RP is effective in CL. This is consistent with previous observations (Kim et al., 2022; Wang et al., 2023).

### 4.2 MULTIPLE-VIEW RANDOM PROJECTION SCHEME

#### 4.2.1 THE MOTIVATION

As shown experimentally in RanPAC and theoretically proven in Section 4.1, projecting feature vectors onto a higher-dimensional space will make representations more separable. Therefore, to improve the model's performance, a naive workaround could be increasing the dimension $d'$ as large as possible. However, this idea is unexpectedly infeasible since it requires inverting a Gram matrix of size $d' \times d'$ (See Section 3). To enable learning in a huge dimensional space and take advantage of the multiple-view learning paradigm for a stronger classifier, we consider $K$ projected spaces with smaller, manageable dimensions (i.e., $K$ single *atomic views*), which all share the same frozen backbone, and each view is generated using a random matrix $\boldsymbol{U}^k, k \in \{1, ..., K\}$.

In what follows, we demonstrate that if we select $k \leq K$ *atomic views* ($d'_a$) and then combine them to produce a *huge view* with a higher dimension (e.g., $d' = kd'_a$), the computation of the original huge view can be significantly simplified via operations on these corresponding atomic views. In particular, let $\boldsymbol{G}, \tilde{\boldsymbol{Z}} \in \mathbb{R}^{N_t \times kd'_a}$, and $\{\tilde{\boldsymbol{Z}}^i\}_{i=1}^k$ (each $\tilde{\boldsymbol{Z}}^i \in \mathbb{R}^{N_t \times d'_a}$) are Gram matrix, representation matrix on a huge view and corresponding atomic views respectively. For each *huge view*, we train a linear expert $\boldsymbol{W} = (\boldsymbol{G} + \lambda \mathbf{I})^{-1} \tilde{\boldsymbol{Z}}^T \boldsymbol{Y}$, where $\tilde{\boldsymbol{Z}} = [\tilde{\boldsymbol{Z}}^i]_{i=1}^k$. We have the theorem below:

**Theorem 4.3.** *If we approximate the Gram matrix* $\boldsymbol{G} = \tilde{\boldsymbol{Z}}^T \tilde{\boldsymbol{Z}}$ *by a block matrix* $\bar{\boldsymbol{G}} = \text{diag}(\tilde{\boldsymbol{Z}}^{i^T} \tilde{\boldsymbol{Z}}^i)_{i=1}^k$ *then the optimal linear classifier on the huge view* $\boldsymbol{W} = [(\boldsymbol{W}^1)^T, ..., (\boldsymbol{W}^k)^T]^T \in \mathbb{R}^{kd'_a \times K_t}$ *where* $\boldsymbol{W}^i = (\boldsymbol{G}^i +$

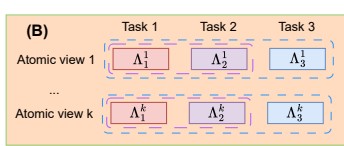

$$p_\nu(x) = \frac{1}{k} \sum_{i=1}^{k} \alpha^i p^i(x)$$

Figure 1: (A) - Multiple-view strategy: in each task, *atomic views* are trained in turn, a $k^{th}$ view is trained to fix the error of the previous one via sample weight $\boldsymbol{\Lambda}^k$. Then, the output of a huge view is represented via the answers of $k$ atomic views, where $\{\alpha^i\}_{i=1}^k$ are auxiliary weights of these views obtained from training. (B) - The process of updating sample weight over CL tasks.

$\lambda \mathbf{I})^{-1} (\tilde{\boldsymbol{Z}}^i)^T \boldsymbol{Y}$ *with* $\boldsymbol{G}^i = (\tilde{\boldsymbol{Z}}^i)^T \tilde{\boldsymbol{Z}}^i \in \mathbb{R}^{d'_a \times K_t}$ *is the corresponding optimal linear classifier on the* $i^{th}$ *atomic view.*

Theorem 4.3 states that under some relaxation, the optimal solution over the huge view can be decomposed into the optimal one for each atomic view. Moreover, in Appendix C.3, we prove that $(\bar{\boldsymbol{G}} + \lambda \mathbf{I})^{-1}$ can approximate well $(\boldsymbol{G} + \lambda \mathbf{I})^{-1}$ when $\lambda$ is sufficiently large. This theorem provides an approximate and effective solution to compute the optimal linear expert for each huge view based on the respective atomic views. Furthermore, the following corollary specifies how the prediction of the linear expert in a huge view can be represented by those of atomic views.

**Corollary 4.4.** *Consider a sample* $\boldsymbol{x}$ *with the representation in the huge view is* $\tilde{\boldsymbol{z}} = [\tilde{\boldsymbol{z}}^i]_{i=1}^k$ *where* $\tilde{\boldsymbol{z}}^i$ *is the representation in the* $i^{th}$ *atomic view. Let* $s^h(\boldsymbol{x}, y) = \tilde{\boldsymbol{z}} \boldsymbol{W}$ *be the huge-view probability prediction vector. Then* $s^h(\boldsymbol{x}, y) = \sum_{i=1}^k s^i(\boldsymbol{x}, y)$ *where* $s^i(\boldsymbol{x}, y)$ *specifies the corresponding response in the* $i^{th}$ *atomic view.*

This corollary provides a convenient way to make a huge-view prediction using atomic-view responses. With $K$ atomic views, we can create up to $2^K - 1$ huge-view answers. We then use these responses in a voting strategy, which will be discussed in the next subsection.

### 4.2.2 Learning Diverse and Complementary Multiple Views via the Boosting Principle

Based on the theoretical arguments in the previous sections, we propose producing the output of the linear expert in each huge view through those of the atomic views. However, we observe that naively training these classifiers separately does not improve the performance as they yield roughly the same outputs. Therefore, creating divergent and complementary linear experts in the atomic views is essential to obtain *an efficient ensemble classifier*.

Our main idea is that as tasks arrive, we continuously maintain $K$ Gram matrices $G_t^{1:K}$ for $K$ atomic views up to task $t$, which can be computed incrementally. Inspired by the AdaBoost principle, we aim for $K$ classifiers on these atomic views to remain supplementary and complementary throughout the training process, even as more tasks are introduced.

We now assume that we have $K$ Gram matrices $G_{t-1}^{1:K}$ for $K$ atomic views up to the task $t-1$ and we need to incrementally build up the Gram matrices $G_t^{1:K}$ when task $t$ arrives in. In particular, we apply our *multiple-view training strategy* to sequentially train a set of linear classifiers for $K$ different atomic views upon the frozen backbone. Inspired by AdaBoost (Freund & Schapire, 1997) and SAMME (Hastie et al., 2009), the $k^{th}$ view ($k > 1$) will be learned to complement the previous one, using the sample weight vector $\mathbf{\Lambda}_t^k$ computed based on the error rate of the $(k-1)^{th}$ view (see Figure 1). We describe this process for each task $t$ in more detail as follows:

**For the first view:**

- First, we start with training the *linear classifier* $g_t^1$ for the first atomic view ($k = 1$) in which all data points possess the same weight of 1 (i.e., $\mathbf{\Lambda}_t^1 = [1, ..., 1]$), following the rule from RanPAC (see Section 3) with the *Gram matrix* $G_t^1 = G_{t-1}^1 + \tilde{\mathbf{Z}}_t^{1^T} \mathrm{diag}(\mathbf{\Lambda}_t^1)\tilde{\mathbf{Z}}_t^1$ where $\tilde{\mathbf{Z}}_t^1$ is the data of task $t$ on the view $k$ through the random projection matrix $\mathbf{U}^1$.

- After that, we compute error rate $err_t^1 = \sum_{i=1}^{n_t} \mathbf{\Lambda}_{t,i}^1 \mathbf{1}_{g_t^1(\mathbf{x}_t^i) \neq y_t^i}$ and auxiliary weight $\alpha_t^1 = \log(\frac{\sum_{i=1}^{n_t} \mathbf{\Lambda}_{t,i}^1 - err_t^1}{err_t^1}) + \log(|\mathcal{Y}_t| - 1)$ for this atomic view where $g_t^1$ is the linear classifier on the first view up to the task $t$.

- We update the sample weight vector w.r.t task $t$, view $k = 2$ as follows:

$$\mathbf{\Lambda}_t^2 = \mathrm{diag}\left(\left[\exp\left\{\alpha_t^1 \mathbf{1}_{g_t^1(\mathbf{x}_t^i) \neq y_t^i}\right\}\right]_{(\mathbf{x}_t^i, y_t^i) \in D_t}\right) \mathbf{\Lambda}_t^1, \tag{2}$$

  where $\mathbf{1}_A$ is the indicator function returning 1 if A is true and 0.

- We normalize the sample weight vector $\mathbf{\Lambda}_t^2$ so that its zero-norm is 1.

**For the $k$-th view, which was learned based on the $(k-1)$-th view:**

- We possess the sample weight vector $\mathbf{\Lambda}_t^k$ for the $k$-th view. We update the *Gram matrix* $G_t^k = G_{t-1}^k + \tilde{\mathbf{Z}}_t^{k^T} \mathrm{diag}(\mathbf{\Lambda}_t^k)\tilde{\mathbf{Z}}_t^k$ which corresponds to the weighted linear classifier $g_t^k$ w.r.t. the sample weights across the tasks $\mathbf{\Lambda}_{1:t}^k$. Note that $\tilde{\mathbf{Z}}_t^k$ represents the data of task $t$ on the view $k$ through the linear random projection $\mathbf{U}^k$.

- Next, we compute error rate $err_t^k = \sum_{i=1}^{n_t} \mathbf{\Lambda}_{t,i}^k \mathbf{1}_{g_t^k(\mathbf{x}_t^i) \neq y_t^i}$ and auxiliary weight $\alpha_t^k = \log(\frac{\sum_{i=1}^{n_t} \mathbf{\Lambda}_{t,i}^k - err_t^k}{err_t^k}) + \log(|\mathcal{Y}_t| - 1)$ for this atomic view.

- We update the sample weight vector w.r.t task $t$, view $k + 1$ as follows:

$$\mathbf{\Lambda}_t^{k+1} = \mathrm{diag}\left(\left[\exp\left\{\alpha_t^k \mathbf{1}_{g_t^k(\mathbf{x}_t^i) \neq y_t^i}\right\}\right]_{(\mathbf{x}_t^i, y_t^i) \in D_t}\right) \mathbf{\Lambda}_t^k. \tag{3}$$

- We normalize the sample weight $\mathbf{\Lambda}_t^{k+1}$ so that its zero-norm is 1.

It is worth noting that given the view $k$, we indeed train the *weighted linear classifier* $g_t^k$ with the sample vectors $\boldsymbol{\Lambda}_{1:t}^k$ on this view across the tasks from 1 to $t$ by solving the following optimization problem (OP):

$$\min_{\boldsymbol{W}^k} \|\text{diag}(\boldsymbol{\Lambda}^k)(\tilde{\boldsymbol{Z}}^k \boldsymbol{W}^k - \boldsymbol{Y})\|_2^2 + \frac{\lambda}{2}\|\boldsymbol{W}^k\|_2^2, \tag{4}$$

where $\boldsymbol{\Lambda}^k = [\boldsymbol{\Lambda}_i^k]_{i=1,...,t}$ and $\tilde{\boldsymbol{Z}}^k = [\tilde{\boldsymbol{Z}}_i^T]_{i=1,...,t}^T \in \mathbb{R}^{N_t \times d_a'}$ (i.e., the stack of data of tasks from 1 to $t$ on the view $k$). The closed-form solution of the OP (4) is given by:

$$\boldsymbol{W}^k = \left((\tilde{\boldsymbol{Z}}^k)^T \text{diag}(\boldsymbol{\Lambda}^k)\tilde{\boldsymbol{Z}}^k + \lambda\mathbf{I}\right)^{-1} \tilde{\boldsymbol{Z}}^k \text{diag}(\boldsymbol{\Lambda}^{k-1})\boldsymbol{Y},$$

where $\tilde{\boldsymbol{Z}}^k$ and $\boldsymbol{Y}$ are respectively features vectors on $k^{th}$ view and labels of data so far. This solution can be updated incrementally across tasks using the Gram matrix $G_t^k$.

After finishing all $T$ tasks, we obtain $K$ sub-classifiers (see Algorithm 2, Appendix D for the overview) and use them in *our proposed voting strategy* to make predictions.

**The proposed voting strategy.** Given a testing example $\boldsymbol{x}$, we first allow the linear classifiers in the atomic views to make their inferences to obtain the prediction probabilities $p^1(\boldsymbol{x}), ..., p^K(\boldsymbol{x})$. The voting strategy is conducted according to the following steps:
❶ Given a threshold $\gamma \in [0,1]$, we select linear classifiers on the atomic views whose confidence level is greater than $\gamma$ (i.e., $\max_y p_y^k(\boldsymbol{x}) \geq \gamma$). Denote this set by $\mathcal{A} \subset \{1, ..., K\}$.
❷ We then create $2^{|\mathcal{A}|} - 1$ *huge views* from $\mathcal{A}$ (where $|\mathcal{A}|$ is the cardinality of $\mathcal{A}$). For each huge view $\mathcal{V} = \{i^1, ..., i^k\} \subset \mathcal{A}$, we compute its prediction probability vector: $p_{\mathcal{V}}(\boldsymbol{x}) = \sum_{k \in \mathcal{V}} \alpha^k p^k(\boldsymbol{x})$, where $\alpha^k = \frac{1}{T}\sum_{t=1}^T \alpha_t^k$.
❸ We carry out majority voting on $2^{|\mathcal{A}|} - 1$ responses of vectors $p_{\mathcal{V}}(\boldsymbol{x})$ to obtain the ultimate label.

In this way, in addition to K weak learners, we have at most $2^K - K - 1$ stronger ones. Then, the probability of obtaining the correct answer will be higher through majority voting.

**Discussion:** It is worth noting that directly applying AdaBoost to CL is challenging because this algorithm requires the data and sample weights of all tasks, which are not always available in CL. Fortunately, thanks to RanPAC's ability to accumulate knowledge from all tasks, we developed an adaptive version of AdaBoost. However, our method is not a simple combination of AdaBoost and RanPAC. In particular, besides adjusting the calculation of the auxiliary weight $\alpha^k$ and normalizing sample weights within the same atomic view $k$ to form $\boldsymbol{\Lambda}^k$, a crucial operation is to produce $2^{|\mathcal{A}|} - 1$ huge views and exploit voting to obtain the final result. This is because the standard version of AdaBoost relies on a large number of weak learners (usually hundreds or thousands or more), which is impractical for working in high-dimensional space according to the design of RanPAC.

### 4.3 TASK-ADAPTIVE BACKBONE AND SELF-IMPROVEMENT PROCESS

This section presents how we improve the backbone of the CP-based method by integrating task-specific prompts and our proposed self-improvement process to select these components properly to achieve higher accuracy.

**Prompt-based task adaptation for CP-backbone.** Since RanPAC only adapts the pre-trained backbone with data from the first task, later task class representations might not be well-separated and could overlap with those of the first task. Therefore, we aim to learn task-adapted representations before applying RP, ensuring distinct separation between different classes and clustering within the same class (see Figure 3, Appendix G.4 for illustration). In this work, we exploit the prompt-based strategy from HiDE (Wang et al., 2023) for training. Specifically, each task $t$ is devoted to a prompt $\boldsymbol{P}_t$ and then trains the current task using Cross-Entropy (CE) loss and a contrastive-based regularization (Reg) loss as follows:

$$\min_{\theta, \boldsymbol{P}_t} CE(\boldsymbol{P}_t, \theta) + \beta Reg(\boldsymbol{P}_t), \tag{5}$$

where $\beta > 0$ is a trade-off parameter and $\theta$ is the weights of temporary classification head of prompted model $f_{\Phi, \boldsymbol{P}}$. Please refer to Appendix E for more details.

**Prompt selection process.** To predict prompts when inference, we learn an auxiliary classification head atop the pre-trained backbone $f_\Phi$. Specifically, given a sample $\boldsymbol{x}$, we use a random matrix $\boldsymbol{U}'$

to project $f_\Phi(x)$ onto a high dimensional space (e.g., $d' = 10,000$) and train a linear classifier $W'$ incrementally like in RanPAC to predict the class labels. Eventually, based on the predicted class label, we can infer the task id $t'$ and the selected prompt $P_{t'}$ for $x$. (See Figure 4A, Appendix G.4).

**Self-improvement process - a two-step prompt selection strategy.** Ideally, the selected prompt $P_{t'}$ for $x$ should coincide with the prompt $P_t$ corresponding to the ground-truth task ID of $x$. However, we observe that $W'$ can sometimes give wrong responses so that $f_{\Phi,P_{t'}}(x)$ can shift much from $f_{\Phi,P_t}(x)$. To further relieve the shift in selecting a wrong prompt, we propose a *two-step prompt prediction* as follows:

❶ First, we use the prompt ID prediction branch based on $f_\Phi(x)$ to predict the prompt $P_{t'}$.
❷ Second, we input $f_{\Phi,P_{t'}}(x)$ into the main class prediction branch to predict the class label and then map it to a task id $t''$ corresponding to the prompt $P_{t''}$.
Finally, we use the prompt $P_{t''}$ to make the final class prediction using $f_{\Phi,P_{t''}}(x)$.

The intuitions behind this *self-improvement process* include (i) the main class prediction branch predicts the class labels and hence the task identity better than the other and (ii) if the main class prediction branch predicts incorrectly the label $y''$ / its task id $t''$ instead of the ground-truth label $y$ and task id $t$ of $x$ (i.e., $y'' \neq y, t'' \neq t$ ), the representations of class $y''$/task $t''$ and $y$/task $t$ tend to stay close. Hence there is a less shift between $f_{\Phi,P_{t''}(x)}$ and $f_{\Phi,P_t(x)}$, leading to less error if using the linear classifier $W$ to predict on $f_{\Phi,P_{t''}(x)}$ rather than $f_{\Phi,P_{t'}(x)}$ (see Figure 4B in Appendix G.4).

## 5 EXPERIMENTS

### 5.1 EXPERIMENTAL SETUP

**Benchmarks.** We examine widely used CIL benchmarks, including Split CIFAR-100, Split ImageNet-R, 5-Datasets, and Split CUB-200 (Please refer Appendix F.1 for more details).

**Baselines and Metrics.** We compare our method with notable methods exploiting pre-trained models for CL scenario, which belong to CP-based approach: RanPAC (McDonnell et al., 2023), ADaM (Zhou et al., 2023), SLCA (Zhang et al., 2023) and prompt-based approach: L2P Wang et al. (2022d), DualPrompt Wang et al. (2022c), S-Prompt++ Wang et al. (2022b), CODA-Prompt Smith et al. (2023), HiDe-Prompt Wang et al. (2023), CPP Li et al. (2024). We present two key metrics: the Final Average Accuracy (FAA), denoting the average accuracy after the last task, and the Final Forgetting Measure (FFM) for all tasks (Appendices F.2 & F.3). The implementation is described in detail in Appendix F.4.

For convenience, we denote two variations of our method without the self-improvement process: (i) *BoostCL-s* is the version that uses the task-adaptive backbone with a single classifier for the CP-based backbone, while (ii) *BoostCL-m(d';K)* is the version that utilizes our proposed ensemble classifier with the boosting strategy for K multiple views each of which has the dimension of d'.

### 5.2 EXPERIMENTAL RESULTS

**Our approach achieves superior results compared to baselines.** Table 1 presents the main results of all the methods. The key observation is that the gap between our method (BoostCL) and the runner-up method is around 2% to 4% in terms of FAA on all considered datasets. Notably, ***with nearly the same number of additional projection parameters***, hence the same high-dimensional space, BoostCL-m (d' = 768; K = 13) still outperforms RanPAC, showing our boosting and voting strategies are beneficial. Furthermore, ***in the absence of Random Projection***, BoostCL-m (w/o RP; K = 15), requiring no additional projection parameters, also surpasses RanPAC about 1% on Split CIFAR-100 and Split ImageNet-R.

Additionally, the results show that BoostCL-s outperform the baselines, and BoostCL-m consistently improves upon this single-view version, indicating the effectiveness of the proposed task-adaptive backbone and multi-view strategies, respectively. Finally, the self-improvement process enhances BoostCL-m's performance by around 1%.

**Time/space complexity of our ensemble classifier**

Table 2 compares our ensemble classifier and RanPAC's classifier on Split CIFAR-100. Firstly, although the classifier of BoostCL-m ($d' = 10,000; K = 15$) requires memory 15 times larger than

Table 1: Overall performance comparison. We provide FAA and FFM of all methods, with standard deviation taken over at least 3 runs of different random seeds. The results corresponding to the best FAA among baselines are underlined.

| Method | Split CIFAR-100 | | Split ImageNet-R | | 5-Datasets | | Split CUB-200 | |
|---|---|---|---|---|---|---|---|---|
| | **FAA** ($\uparrow$) | FFM ($\downarrow$) | **FAA** ($\uparrow$) | FFM ($\downarrow$) | **FAA** ($\uparrow$) | FFM ($\downarrow$) | **FAA** ($\uparrow$) | FFM ($\downarrow$) |
| L2P | 83.06 ±0.17 | 6.58 ±0.40 | 63.65 ±0.12 | 7.51 ±0.17 | 81.84 ±0.95 | 4.58 ±0.53 | 74.52 ±0.92 | 11.25 ±0.23 |
| DualPrompt | 86.60 ±0.19 | 4.45 ±0.16 | 68.79 ±0.31 | 4.49 ±0.14 | 77.91 ±0.45 | 13.17 ±0.71 | 82.05±0.95 | 3.56 ±0.53 |
| S-Prompt++ | 88.81 ±0.18 | 3.87 ±0.05 | 69.68 ±0.12 | 3.29 ±0.05 | 86.19 ±0.65 | 4.67 ±0.72 | 83.12 ±0.54 | 2.72 ±0.64 |
| CODA-P | 86.94 ±0.63 | 4.04 ±0.18 | 70.03 ±0.47 | 5.17 ±0.22 | 64.20 ±0.53 | 17.22 ±0.55 | 74.34 ±0.68 | 12.05 ±0.41 |
| HiDe-Prompt | 92.61 ±0.28 | 3.16 ±0.10 | 75.06 ±0.12 | **2.17** ±0.19 | 93.92 ±0.33 | 0.31 ±0.12 | 86.62 ±0.35 | **1.98** ±0.15 |
| CPP | 91.12 ±0.12 | 3.33 ±0.18 | 74.88 ±0.07 | 3.65 ±0.03 | 92.92 ±0.17 | **0.19** ±0.07 | 82.35 ±0.23 | 3.24 ±0.32 |
| ADaM | 87.60 | - | 72.30 | - | 74.15 | - | 87.10 | - |
| SLCA | 91.50 | - | 77.00 | - | 84.71 | - | 84.70 | - |
| RanPAC | 92.20 | - | 77.90 | - | 82.85 | - | 90.30 | - |
| BoostCL-s (d'=10k; K=1) | 94.03 ±0.23 | 1.82 ±0.12 | 78.52 ±0.12 | 3.12 ±0.15 | 94.08 ±0.25 | 0.25 ±0.15 | 90.92 ±0.32 | 2.21 ±0.11 |
| BoostCL-m (d'=768; K=13) | 94.56±0.23 | 1.75 ±0.12 | 78.65 ±0.12 | 3.12 ±0.15 | 94.66±0.41 | 0.23±0.28 | 91.25±0.52 | 2.12±0.19 |
| BoostCL-m (w/o RP; K=13) | 94.65 ±0.25 | 1.71 ±0.15 | 78.82 ±0.10 | 3.03 ±0.15 | 94.96 ±0.14 | 0.25±0.33 | 91.60±0.46 | 2.12 ±0.21 |
| BoostCL-m (d'=10k; K=15) | 95.45 ±0.25 | 1.72 ±0.15 | 79.62 ±0.10 | 3.03 ±0.15 | 95.62 ±0.30 | 0.22 ±0.13 | 92.05 ±0.31 | 2.08 ±0.12 |
| BoostCL | **96.55** ±0.32 | **1.15** ±2.12 | **80.42** ±0.15 | 3.05 ±0.12 | **96.73** ±0.13 | 0.30 ±0.23 | **93.03** ±0.32 | 2.13 ±0.15 |

RanPAC ($d' = 10,000$), this expense can be worth compensating because our BoostCL-m achieves a higher FAA by a margin of more than 3%. Additionally, considering BoostCL ($d' = 768; K = 13$), which requires less memory than RanPAC, we can see that with the help of our ensemble strategy, it can still improve the FAA compared to RanPAC and even BoostCL's single view ($d' = 10,000$). We also conduct experiments on the model using RanPAC's backbone adapting with our ensemble classifier *(RanPAC++)*. The results once again confirm the effectiveness of our ensemble classifier. Furthermore, in the case where RanPAC uses $d' = 10,000 \times 15 = 150,000$ *(RanPAC\*)*, which is the same as the total dimensions in BoostCL, the cost to store the Gram matrix (with size $d' \times d'$) or its inversion is too high and impractical. Meanwhile, BoostCL works well with this setting, as it only needs to deal with submatrices.

Regrading forward time, comparing with the original RanPAC ($d' = 10,000$), forwarding the classifier of BoostCL ($d' = 10,000; K = 15$) requires more time. This is obvious because we use an ensemble strategy for prediction instead of using only one classification head like RanPAC. This cost is offset by the performance improvement we achieve. In addition, when discussing the potential of using projection matrices on high-dimensional space, we claimed in our paper that our process helps to alleviate the "substantial computational cost when computing the inverse of a $d' \times d'$ Gram matrix for solving the linear classifier" which is done in RanPAC. In this table, we compare our BoostCL with RanPAC$*$ ($d' = 10,000 \times 15$) and mark that this version of RanPAC is infeasible to train because the cost when computing the inversion of a Gram matrix of size $d'(= 10,000 \times 15)$ is $O(d'^c)$ for some constant $c \in [2.3, 3.0]$, according to Geéron (2017). Therefore, BoostCL has solved this computational crisis, saving the time cost for a factor of $O(15^c)$. Training time for our methods and baselines is reported in Appendix G.3.

Here we note that the times reported in Table 2 are *the forward time* of the classification head. The *training times* of our approach and baselines are reported in Appendix G.3. We also provide additional experimental results to scrutinize other aspects of our multiple-view strategy, including the effects of the total number of atomic views and the dimension of each corresponding projected space, the role of our voting strategy, the applicability of our ensemble classifier (Appendix G.1) and the time/storage complexity (Appendix G.3).

**Effect of task-adaptive backbone**

The results in Table 1, with the outperformance of BoostCL-s over CP-based baselines, indicate that employing task-specific prompts to adapt to CL tasks is beneficial, confirming our argument.

Table 2: Computation and storage cost of **the classification head** of methods, on Split CIFAR-100.

| Method | Metric | | | |
|---|---|---|---|---|
| | # Params | Forward time (s) | FAA | Can train? |
| RanPAC ($d' = 10,000$) | $100 \times 10^4$ | $1.77 \times 10^{-4} \pm 2.58 \times 10^{-5}$ | 92.20 | Yes |
| BoostCL ($d' = 10,000; K = 1$) | $100 \times 10^4$ | $1.77 \times 10^{-4} \pm 2.58 \times 10^{-5}$ | 94.03 | Yes |
| BoostCL ($d' = 768; K = 13$) | less than $100 \times 10^4$ | $8.91 \times 10^{-4} \pm 2.35 \times 10^{-5}$ | 94.56 | Yes |
| RanPAC* ($d' = 10,000 \times 15$) | $100 \times 15 \times 10^4$ | $2.07 \times 10^{-3} \pm 2.65 \times 10^{-4}$ | - | **No** |
| BoostCL ($d' = 10,000; K = 15$) | $100 \times 15 \times 10^4$ | $3.31 \times 10^{-3} \pm 4.26 \times 10^{-4}$ | 95.45 | Yes |
| BoostCL (get 15 outputs) | | $2.68 \times 10^{-3} \pm 3.65 \times 10^{-4}$ | | |
| BoostCL (voting) | | $6.25 \times 10^{-4} \pm 6.31 \times 10^{-5}$ | | |

Additionally, Figure 2, depicting the latent space of RanPAC's and our model on Split CIFAR-100, supports these results by showcasing better clustering of classes with more adapted tasks.

**Effect of self-improvement process**

Employing multiple steps for prompt prediction enhances model accuracy in general, as evident in Table 1, where 2 steps improve FAA by around 1% across all datasets. In addition, Table 3 provides a detailed perspective, showing that as the number of steps increases, model performance further improves. The most significant improvement is observed when transitioning from 1 step to 2 steps, with increases of 1.1% on Split CIFAR-100 and 0.8% on Split ImageNet-R. This

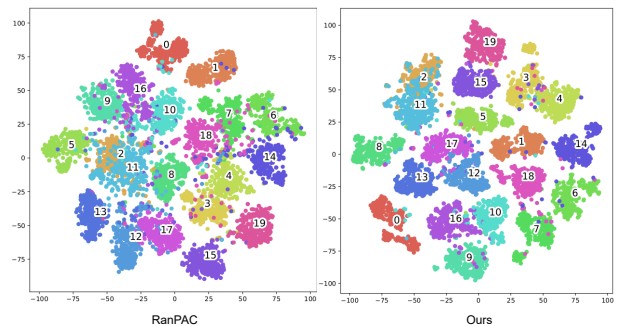

Figure 2: t-SNE visualization for Split CIFAR-100 on data of Task 1 (class $0 \to 9$) and Task 2 (class $10 \to 19$).

improvement tends to gradually converge in subsequent steps, with growths of less than 0.2% from step 4 to step 5 in both datasets. This limitation may stem from the simplicity of the method itself; however, it could serve as an interesting suggestion for future research.

Table 3: Performance when increasing the number of prompt prediction steps

| Benchmark | Number of steps | | | | |
|---|---|---|---|---|---|
| | 1 | 2 | 3 | 4 | 5 |
| Split CIFAR-100 | 95.45 | 96.55 | 96.85 | 97.03 | 97.05 |
| Split ImageNet-R | 79.62 | 80.42 | 80.62 | 80.65 | 80.66 |

## 6    CONCLUSION

In this work, we first consider the role and implications of a technique that is experimentally effective in previous work, using random projection for pre-trained models in CL. Then, the benefits of RP in high-dimensional space motivated us to embark on a journey of theoretical thinking and propose a multi-view strategy for an efficient classifier, in which the principle of Adaboost is adapted to overcome inherent obstacles and applied for the first time in CL. In addition, our self-improvement process technique, although simple, also shows significant effectiveness in selecting proper task-specific prompts. The experimental results demonstrate a positive impact of our proposed method in improving model quality while only applying to linear classifiers.

**Limitations:** A primary limitation lies in **the computational and storage costs for our ensemble classifier**. Like other ensemble methods, we need to train and store many weak learners than usual (see Appendix G.3). However, thanks to the voting strategy, we do not need to train as many weak learners to achieve significant performance as the original methods (Adaboost and SAMME often need hundreds of weak learners to show improvement). Another potential limitation can be **the computational overhead of the self-improvement process**. As we discussed computational complexity in Section G.3.3, two rounds of this process can already bring improvement, and its time complexity is not too significant, so it could be used with any prompt-based methods. Nevertheless, in cases where downstream data is too different from that of the pre-trained-ViT, the simple design of the prompt predictor may hinder its performance. Consequently, adding more self-improvement steps may not increase accuracy.

## ACKNOWLEDGMENT

Trung Le and Dinh Phung were supported by ARC DP23 grant DP230101176 and by the Air Force Office of Scientific Research under award number FA2386-23-1-4044.

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

# Appendices

## A WHY DOES RANDOM PROJECTION ONTO HIGHER DIMENSION SPACE HELP INCREASE THE MARGIN?

We first consider the role of RP without activation function. Let $\mathcal{H}$ be the set of all measurable functions that map from $\mathbb{R}^d$ to $\mathcal{Y}$, and $\mathcal{H}'$ be the set of all measurable functions that map from $\mathbb{R}^{d'}$ to $\mathcal{Y}$. The following theorem shows the benefit of random projection onto a high-dimensional space.

**Theorem A.1.** *Let $U$ be a random matrix of size $d' \times d$, with $d' \gg d$, whose elements are independent copies of a Gaussian random variable with unit variance and 0 mean. Consider $s = (z, y)$ and its projection $s^u = (z^T U, y)$. For every $\epsilon > 0$, with probability at least $1 - (C.\epsilon)^{d'-d+1} - e^{-c.d'}$ where $C$ and $c$ are some constants, we have*

$$\gamma_{in}(S^u, \mathcal{H}') \geq \gamma_{in}(S, \mathcal{H})(\sqrt{d'} - \sqrt{d-1})\epsilon$$

This theorem suggests that the margin in the new space after applying RP should increase fast as $d'$ increases. The increase rate is $O(\sqrt{d'})$. This property of RP is intriguing and has never been known before. Note that such an increase may not appear in the cases of class overlapping due to $\gamma_{\text{in}}(s, \mathcal{H}) = 0$ for some sample $s$. Therefore, RP will provide a crucial benefit in the cases of (nonlinear or linear) separability.

In practice, RP is often followed by a fixed nonlinear operation. One main reason is that this nonlinearity can keep the crucial benefits of RP when used before, e.g., linear models or neural networks. Without nonlinearity, the effect of RP can be absorbed into a trainable linear operation. Therefore, we need to incorporate nonlinearity into the analysis of RP to explain the overall benefits in practice.

In general, nonlinearity often poses challenges for analyzing RP. We restrict our analysis to nonlinear operations which are *invertible* and *expansive*.[2] Note that such invertible and expansive nonlinearities are prevalent, such as $\sigma(x) = bx^2$ in domain $[1, \infty)$ for any $b \geq 1$, $\sigma(x) = be^x$ in domain $[0, \infty)$, $\sigma(x) = b\sin(x)$ in domain $[-a, a]$ for some small $a > 0$, $\sigma(x) = b\ln(x)$ in domain $(0, 1)$, $\sigma(x) = bx$, $LeakyReLU(x) = \max(0, x) + b\min(0, x)$, etc,. This leads to our Theorem 4.2

### A.1 MARGIN AND GENERALIZATION ERROR

We next recall a well-known connection (Bartlett et al., 2017; Sokolić et al., 2017; Xu & Mannor, 2012) between the margin and generalization ability of a classifier.

Consider a *learning problem* specified by a model (or hypothesis) class $\mathcal{H}$, an instance set $\mathcal{X}$, and a loss function $\ell : \mathcal{H} \times \mathcal{X} \to \mathbb{R}_{\geq 0}$, where each input instance $x \in \mathcal{X}$ has a corresponding output $y \in \mathcal{Y}$. However, without loss of generality, we omit the output $y$ for simplicity of presentation. Given a distribution $P$ defined on $\mathcal{X}$, the quality of a model $h \in \mathcal{H}$ is measured by its *expected loss* $F(P, h) = \mathbb{E}_{x \sim P}[\ell(h, x)]$. The *empirical loss* $F(S, h) = \frac{1}{|S|}\sum_{x \in S} \ell(h, x)$ is defined from a finite set $S = \{x_1, ..., x_n\} \subseteq \mathcal{X}$ of size $n$.

Let $\Gamma(\mathcal{X}) := \bigcup_{i=1}^K \mathcal{X}_i$ be a partition of $\mathcal{X}$ into $K$ disjoint nonempty subsets. Denote $S_i = S \cap \mathcal{X}_i$, and $n_i = |S_i|$ as the number of samples falling into $\mathcal{X}_i$, meaning that $n = \sum_{j=1}^K n_j$. Denote $T = \{i \in [K] : S \cap \mathcal{X}_i \neq \emptyset\}$.

The following result is a simple consequence from algorithmic robustness (Kawaguchi et al., 2022; Xu & Mannor, 2012).

**Theorem A.2.** *Consider a model $h$ and a dataset $S$ which consists of $n$ i.i.d. samples from distribution $P$. Let $\ell(h, x)$ be the loss of $h$ at instance $x$, and $C_h = \sup_{x \in \mathcal{X}} \ell(h, x)$, $\epsilon(S) = \sup_{i \in [K]} \sup_{x \in \mathcal{X}_i, s \in S_i} |\ell(h, x) - \ell(h, s)|$. For any $\delta > 0$, the following holds with probability at least $1 - \delta$:*

$$F(P, h) \leq F(S, h) + \epsilon(S) + 3C_h\sqrt{\frac{|T|\ln(2K/\delta)}{n}} + \frac{2C_h|T|\ln(2K/\delta)}{n} \tag{6}$$

---

[2]A function $\sigma$ is expansive if it satisfies $|\sigma(x) - \sigma(y)| \geq |x - y|$ for all $x, y$ in their domain.

This theorem implies that the expected loss of a model can be bounded by using robustness level $\epsilon(\boldsymbol{S})$ of the model around the training samples. A more robust model can suggest better generalization on unseen data.

Next we connect the margin and generalization ability of a model. Let $\mathcal{N}(\mathcal{X}, \|\cdot\|, \lambda)$ be the $\lambda$-*covering number* of $\mathcal{X}$ according to some metric $\|\cdot\|$ and $\Gamma(\mathcal{X})$ be the corresponding covering of $\mathcal{X}$, where $\lambda$ is a positive constant. This means $K = \mathcal{N}(\mathcal{X}, \|\cdot\|, \lambda)$. When the input space $\mathcal{X} \subseteq \mathbb{R}^d$ has diameter at most $R$, a classical fact (Wu & Yang, 2016) says that $\mathcal{N}(\mathcal{X}, \|\cdot\|, \lambda) \leq \left(\frac{2R\sqrt{d}}{\lambda}\right)^d$. Furthermore, Example 1 in (Xu & Mannor, 2012) shows that $\epsilon(\boldsymbol{S}) = 0$ for any $\lambda \leq 0.5\gamma_h$ and 0-1 loss $\ell$, where $\gamma_h = \max\{v : \|\boldsymbol{x} - \boldsymbol{s}\| \leq v \Rightarrow h(\boldsymbol{x}) = h(\boldsymbol{s})$ for all $\boldsymbol{x} \in \mathcal{X}, \boldsymbol{s} \in \boldsymbol{S}\}$ denotes the margin of $\boldsymbol{h}$ for the training set. Combining those observations with Theorem A.2 will lead to the following.

**Corollary A.3.** *Given the notations in Theorem A.2 with 0-1 loss $\ell$ and $\mathcal{X} \subseteq \mathbb{R}^d$ with diameter at most $R$. Denote $\gamma_h = \max\{v : \|\boldsymbol{x} - \boldsymbol{s}\| \leq v \Rightarrow h(\boldsymbol{x}) = h(\boldsymbol{s})$ for all $\boldsymbol{x} \in \mathcal{X}, \boldsymbol{s} \in \boldsymbol{S}\}$ as the margin of $\boldsymbol{h}$, and $A = \ln(2/\delta) + d\ln(4R\sqrt{d}/\gamma_h)$. If $\gamma_h > 0$, then the following holds with probability at least $1 - \delta$:*

$$F(P, \boldsymbol{h}) \leq F(\boldsymbol{S}, \boldsymbol{h}) + 3\sqrt{\frac{|\boldsymbol{T}|A}{n}} + \frac{2|\boldsymbol{T}|A}{n} \tag{7}$$

This result suggests that a model with larger margin can lead to smaller bound for the expected loss, suggesting better generalization ability. Combining this with the RP's ability to increase margin, we can conclude that RP can facilitate better generalization ability of a model in high dimensional spaces.

### A.1.1 PROOF OF THEOREM A.2

*Proof.* Firstly, we make the following decomposition:

$F(P, \boldsymbol{h})$

$$= F(P, \boldsymbol{h}) - \sum_{i=1}^{K} \frac{n_i}{n} \mathbb{E}_{\boldsymbol{x}}[f(\boldsymbol{h}, \boldsymbol{x})|\boldsymbol{x} \in \mathcal{X}_i] + \sum_{i=1}^{K} \frac{n_i}{n} \mathbb{E}_{\boldsymbol{x}}[f(\boldsymbol{h}, \boldsymbol{x})|\boldsymbol{x} \in \mathcal{X}_i] - F(\boldsymbol{S}, \boldsymbol{h}) + F(\boldsymbol{S}, \boldsymbol{h}) \tag{8}$$

Secondly, consider $A_1 = F(P, \boldsymbol{h}) - \sum_{i=1}^{K} \frac{n_i}{n} \mathbb{E}_{\boldsymbol{x}}[f(\boldsymbol{h}, \boldsymbol{x})|\boldsymbol{x} \in \mathcal{X}_i]$. Observe that:

$$A_1 = F(P, \boldsymbol{h}) - \sum_{i=1}^{K} \frac{n_i}{n} \mathbb{E}_{\boldsymbol{x}}[f(\boldsymbol{h}, \boldsymbol{x})|\boldsymbol{x} \in \mathcal{X}_i] \tag{9}$$

$$= \mathbb{E}_{\boldsymbol{x}}[f(\boldsymbol{h}, \boldsymbol{x})] - \sum_{i=1}^{K} \frac{n_i}{n} \mathbb{E}_{\boldsymbol{x}}[f(\boldsymbol{h}, \boldsymbol{x})|\boldsymbol{x} \in \mathcal{X}_i] \tag{10}$$

Note that $(n_1, ..., n_K)$ is an i.i.d multinomial random variable with parameters $n$ and $(P(\mathcal{X}_1), ..., P(\mathcal{X}_K))$. Therefore

$$A_1 = \sum_{i=1}^{K} P(\mathcal{X}_i)\mathbb{E}_{\boldsymbol{x}}[f(\boldsymbol{h}, \boldsymbol{x})|\boldsymbol{x} \in \mathcal{X}_i] - \sum_{i=1}^{K} \frac{n_i}{n} \mathbb{E}_{\boldsymbol{x}}[f(\boldsymbol{h}, \boldsymbol{x})|\boldsymbol{x} \in \mathcal{X}_i] \tag{11}$$

$$= \sum_{i=1}^{K} \mathbb{E}_{\boldsymbol{x}}[f(\boldsymbol{h}, \boldsymbol{x})|\boldsymbol{x} \in \mathcal{X}_i] \left[P(\mathcal{X}_i) - \frac{n_i}{n}\right] \tag{12}$$

$$\leq Q\sqrt{\frac{|\boldsymbol{T}|\log(2K/\delta)}{n}} + a_c \frac{2|\boldsymbol{T}|\log(2K/\delta)}{n} \tag{13}$$

We have the last inequality with probability at least $1 - \delta$, according to Theorem 3 in (Kawaguchi et al., 2022), where $Q = a_t\sqrt{2} + a_c, a_t = \max_{i \in \boldsymbol{T}} \mathbb{E}_{\boldsymbol{x}}[f(\boldsymbol{h}, \boldsymbol{x})|\boldsymbol{x} \in \mathcal{X}_i]$, and $a_c = \max_{j \notin \boldsymbol{T}} \mathbb{E}_{\boldsymbol{x}}[f(\boldsymbol{h}, \boldsymbol{x})|\boldsymbol{x} \in \mathcal{X}_j]$. Note that $\mathbb{E}_{\boldsymbol{x}}[f(\boldsymbol{h}, \boldsymbol{x})|\boldsymbol{x} \in \mathcal{X}_i] \leq C_h$ for any index $i$. It suggests that $Q \leq 3C_h$ and $a_c \leq C_h$. As a result,

$$A_1 \leq 3C_h\sqrt{\frac{|\boldsymbol{T}|\log(2K/\delta)}{n}} + \frac{2C_h|\boldsymbol{T}|\log(2K/\delta)}{n} \tag{14}$$

Third, consider $A_2 = \sum_{i=1}^{K} \frac{n_i}{n} \mathbb{E}_{\boldsymbol{x}}[\ell(\boldsymbol{h}, \boldsymbol{x}) | \boldsymbol{x} \in \mathcal{X}_i] - F(\boldsymbol{S}, \boldsymbol{h})$. Observe that

$$
\begin{aligned}
A_2 &= \sum_{i=1}^{K} \frac{n_i}{n} \mathbb{E}_{\boldsymbol{x}}[\ell(\boldsymbol{h}, \boldsymbol{x}) | \boldsymbol{x} \in \mathcal{X}_i] - \frac{1}{n} \sum_{\boldsymbol{s} \in \boldsymbol{S}} \ell(\boldsymbol{h}, \boldsymbol{s}) \\
&= \sum_{i \in \boldsymbol{T}} \frac{n_i}{n} \mathbb{E}_{\boldsymbol{x}}[\ell(\boldsymbol{h}, \boldsymbol{x}) | \boldsymbol{x} \in \mathcal{X}_i] - \frac{1}{n} \sum_{i \in \boldsymbol{T}} \sum_{\boldsymbol{s} \in \boldsymbol{S}_i} \ell(\boldsymbol{h}, \boldsymbol{s}) \\
&= \frac{1}{n} \sum_{i \in \boldsymbol{T}} \left( n_i \mathbb{E}_{\boldsymbol{x}}[\ell(\boldsymbol{h}, \boldsymbol{x}) | \boldsymbol{x} \in \mathcal{X}_i] - \sum_{\boldsymbol{s} \in \boldsymbol{S}_i} \ell(\boldsymbol{h}, \boldsymbol{s}) \right) \\
&= \frac{1}{n} \sum_{i \in \boldsymbol{T}} \sum_{\boldsymbol{s} \in \boldsymbol{S}_i} \left( \mathbb{E}_{\boldsymbol{x}}[\ell(\boldsymbol{h}, \boldsymbol{x}) | \boldsymbol{x} \in \mathcal{X}_i] - \ell(\boldsymbol{h}, \boldsymbol{s}) \right) \\
&= \frac{1}{n} \sum_{i \in \boldsymbol{T}} \sum_{\boldsymbol{s} \in \boldsymbol{S}_i} \mathbb{E}_{\boldsymbol{x}} \left[ \ell(\boldsymbol{h}, \boldsymbol{x}) - \ell(\boldsymbol{h}, \boldsymbol{s}) | \boldsymbol{x} \in \mathcal{X}_i \right] \\
&\leq \frac{1}{n} \sum_{i \in \boldsymbol{T}} \sum_{\boldsymbol{s} \in \boldsymbol{S}_i} \sup_{\boldsymbol{x} \in \mathcal{X}_i} |\ell(\boldsymbol{h}, \boldsymbol{x}) - \ell(\boldsymbol{h}, \boldsymbol{s})| \\
&\leq \frac{1}{n} \sum_{i \in \boldsymbol{T}} \sum_{\boldsymbol{s} \in \boldsymbol{S}_i} \epsilon(\boldsymbol{S}) \\
&= \epsilon(\boldsymbol{S}) 
\end{aligned}
\tag{15}
$$

Finally, decomposition (8) shows that $F(P, \boldsymbol{h}) = A_1 + A_2$. Combining this with (14), and (15) will complete the proof. $\qquad \square$

## A.2 PROOF OF THEOREM A.1

Let $s \in \boldsymbol{S}$ be an instance and $g_h \in \mathcal{H}$ be the classifier such that $\gamma(s, g_h, \mathbb{R}^d) = \gamma_{\text{in}}(s, \mathcal{H})$. We first consider the ball that reflects the region around instance $s = (\boldsymbol{z}, y)$, for which the classifier $g_h$ has no misclassification. Denoting $\nu = \gamma_{\text{in}}(s, \mathcal{H})$, the ball is:

$$
\mathbb{B}(s, \nu) = \{\boldsymbol{z}' \in \mathbb{R}^d : \|\boldsymbol{z} - \boldsymbol{z}'\| \leq \nu, g_h(\boldsymbol{z}') = y\}
\tag{16}
$$

The diameter of this ball represents the margin of the classifier at sample $s$. Every point in this ball will be assigned the same label with $\boldsymbol{z}$. When projecting this ball onto a higher-dimensional space by transform matrix $\boldsymbol{U}$ of size $d \times d'$, we obtain an ellipse as

$$
\mathbb{B}^u(s, \nu) = \{\boldsymbol{z}'^T \boldsymbol{U} : \boldsymbol{z}' \in \mathbb{B}(s, \nu)\}
\tag{17}
$$

Denote $g'(\boldsymbol{v}) = g_h((\boldsymbol{U}\boldsymbol{U}^T)^{-1} \boldsymbol{U}\boldsymbol{v})$ for any $\boldsymbol{v} \in \mathbb{R}^{d'}$. Note that $g'$ is a classifier, defined in the new space, and hence $g' \in \mathcal{H}'$. This function almost surely exists since $\boldsymbol{U}\boldsymbol{U}^T$ is invertible, almost surely Rudelson & Vershynin (2009). Observe the margin of $s^u = (\boldsymbol{z}^T \boldsymbol{U}, y)$ w.r.t $g'$:

$$
\begin{aligned}
\gamma(s^u, g', \mathbb{R}^{d'}) &= \sup\{\mu : \|\boldsymbol{z}^T \boldsymbol{U} - \boldsymbol{v}\| \leq \mu \Rightarrow g'(\boldsymbol{v}) = y, \forall \boldsymbol{v} \in \mathbb{R}^{d'}\} & (18) \\
&= \sup\{\mu : \|\boldsymbol{z}^T \boldsymbol{U} - \boldsymbol{v}\| \leq \mu \Rightarrow g_h((\boldsymbol{U}\boldsymbol{U}^T)^{-1} \boldsymbol{U}\boldsymbol{v}) = y, \forall \boldsymbol{v} \in \mathbb{R}^{d'}\} & (19) \\
&= \sup\{\mu : \|\boldsymbol{z}^T \boldsymbol{U} - \boldsymbol{z}'^T \boldsymbol{U}\| \leq \mu \Rightarrow g_h(\boldsymbol{z}') = y, \forall \boldsymbol{z}' \in \mathbb{R}^d\} & (20) \\
&= \sup\{\mu : \|(\boldsymbol{z} - \boldsymbol{z}')^T \boldsymbol{U}\| \leq \mu \Rightarrow g_h(\boldsymbol{z}') = y, \forall \boldsymbol{z}' \in \mathbb{R}^d\} & (21)
\end{aligned}
$$

This suggests that $\gamma(s^u, g', \mathbb{R}^{d'})$ represents the smallest diameter of $\mathbb{B}^u(s, \nu)$. Furthermore, the smallest diameter of this ellipse $\mathbb{B}^u$ is, in fact, the smallest singular value of matrix $\nu\boldsymbol{U}$. It means

$$
\gamma(s^u, g', \mathbb{R}^{d'}) = \sigma_{min}(\nu\boldsymbol{U}) = \nu\sigma_{min}(\boldsymbol{U})
\tag{22}
$$

where $\sigma_{min}(\boldsymbol{U})$ denotes the smallest singular value of matrix $\boldsymbol{U}$.

Remember that $U$ is a random matrix whose elements are independent copies of a Gaussian random variable with unit variance and zero mean. According to Theorem 1.1 in (Rudelson & Vershynin, 2009), for every $\epsilon > 0$, we have

$$\Pr(\sigma_{min}(U) < \epsilon[\sqrt{d'} - \sqrt{d-1}]) \leq (C.\epsilon)^{d'-d+1} + e^{-c.d'}$$

for some constants $C$ and $c$. This implies that

$$\Pr(\sigma_{min}(U) \geq \epsilon[\sqrt{d'} - \sqrt{d-1}]) > 1 - (C.\epsilon)^{d'-d+1} - e^{-c.d'}$$

As a consequence

$$\Pr(\gamma(s^u, g', \mathbb{R}^{d'}) \geq \nu\epsilon[\sqrt{d'} - \sqrt{d-1}]) > 1 - (C.\epsilon)^{d'-d+1} - e^{-c.d'}$$

Note that, by definition, the Bayes margin $\gamma_{\text{in}}(s^u, \mathcal{H}')$ cannot be less than $\gamma(s^u, g', \mathbb{R}^{d'})$ due to $g' \in \mathcal{H}'$. Therefore

$$\Pr(\gamma_{\text{in}}(s^u, \mathcal{H}') \geq \nu\epsilon[\sqrt{d'} - \sqrt{d-1}]) > 1 - (C.\epsilon)^{d'-d+1} - e^{-c.d'}$$

The margin of each instance is enlarged by a factor of $\epsilon[\sqrt{d'} - \sqrt{d-1}])$. Therefore $\gamma(S, g_h, \mathbb{R}^d) = \min_{s \in S} \gamma(s, g_h, \mathbb{R}^d)$ is also enlarged by the same factor, completing the proof.

## A.3 Proof of Theorem 4.2

Let $s \in S$ be an instance and $g_h \in \mathcal{H}$ be the classifier such that $\gamma(s, g_h, \mathbb{R}^d) = \gamma_{\text{in}}(s, \mathcal{H})$. We first consider the ball that reflects the region around instance $s = (z, y)$, for which the classifier $g_h$ does not have any misclassification. Denoting $\nu = \gamma_{\text{in}}(s, \mathcal{H})$, the ball is:

$$\mathbb{B}(s, \nu) = \{z' \in \mathbb{R}^d : \|z - z'\| \leq \nu, g_h(z') = y\} \tag{23}$$
$$= \{z' \in \mathbb{R}^d : \|z - z'\| \leq \nu, g_h(z') = g_h(z)\} \tag{24}$$

The diameter of this ball represents the margin of the classifier at sample $s$. Every point in this ball will be assigned the same label with $z$. When projecting this ball onto a higher-dimensional space by transform matrix $U$ of size $d \times d'$, we obtain the following ellipse:

$$\mathbb{B}^u(s, \nu) = \{z'^T U : z' \in \mathbb{B}(s, \nu)\} \tag{25}$$

Denote $g'(v) = g_h((UU^T)^{-1}U\sigma^{-1}(v))$ for any $v \in \mathbb{R}^{d'}$, where $\sigma^{-1}$ denotes the inverse function of $\sigma$. Note that $g'$ is a classifier, defined in the new space, and hence $g' \in \mathcal{H}'$. This function almost surely exists since $UU^T$ is invertible, almost surely Rudelson & Vershynin (2009). Observe the margin of $s^u = (\sigma(z^T U), y)$ w.r.t $g'$:

$$\gamma(s^u, g', \mathbb{R}^{d'}) = \sup\{\mu : \|\sigma(z^T U) - v\| \leq \mu \Rightarrow g'(v) = y, \forall v \in \mathbb{R}^{d'}\} \tag{26}$$
$$= \sup\{\mu : \|\sigma(z^T U) - v\| \leq \mu \Rightarrow g_h((UU^T)^{-1}U\sigma^{-1}(v)) = g_h(z), \forall v\} \tag{27}$$
$$= \sup\{\mu : \|\sigma(z^T U) - \sigma(z'^T U)\| \leq \mu \Rightarrow g_h(z') = g_h(z), \forall z' \in \mathbb{R}^d\} \tag{28}$$

The last equality comes from the fact that $\sigma$ is an one-to-one mapping which maps each $v \in \mathbb{R}^{d'}$ to $\sigma(z'^T U)$, where $z' = (UU^T)^{-1}U\sigma^{-1}(v)$.

Since $\sigma$ is an expansive function, we have $\|\sigma(z^T U) - \sigma(z'^T U)\| \geq \|z^T U - z'^T U\| = \|(z - z')^T U\|$. Combining this fact with (28) implies that $\gamma(s^u, g', \mathbb{R}^{d'})$ is not less than the smallest diameter of $\mathbb{B}^u(s, \nu)$. Furthermore, the smallest diameter of $\mathbb{B}^u$ is, in fact, the smallest singular value of matrix $\nu U$. It means

$$\gamma(s^u, g', \mathbb{R}^{d'}) \geq \sigma_{min}(\nu U) = \nu\sigma_{min}(U) \tag{29}$$

where $\sigma_{min}(U)$ denotes the smallest singular value of matrix $U$.

Remember that $U$ is a random matrix whose elements are independent copies of a Gaussian random variable with unit variance and zero mean. According to Theorem 1.1 in (Rudelson & Vershynin, 2009), for every $\epsilon > 0$, we have

$$\Pr(\sigma_{min}(U) < \epsilon[\sqrt{d'} - \sqrt{d-1}]) \leq (C.\epsilon)^{d'-d+1} + e^{-c.d'}$$

for some constants $C$ and $c$. This implies that

$$\Pr(\sigma_{min}(\boldsymbol{U}) \geq \epsilon[\sqrt{d'} - \sqrt{d-1}]) > 1 - (C.\epsilon)^{d'-d+1} - e^{-c.d'}$$

As a consequence

$$\Pr(\gamma(s^u, g', \mathbb{R}^{d'}) \geq \nu\epsilon[\sqrt{d'} - \sqrt{d-1}]) > 1 - (C.\epsilon)^{d'-d+1} - e^{-c.d'}$$

Note that, by definition, the Bayes margin $\gamma_{\text{in}}(s^u, \mathcal{H}')$ cannot be less than $\gamma(s^u, g', \mathbb{R}^{d'})$ due to $g' \in \mathcal{H}'$. Therefore

$$\Pr(\gamma_{\text{in}}(s^u, \mathcal{H}') \geq \nu\epsilon[\sqrt{d'} - \sqrt{d-1}]) > 1 - (C.\epsilon)^{d'-d+1} - e^{-c.d'}$$

The margin of each instance is enlarged by a factor of $\epsilon[\sqrt{d'} - \sqrt{d-1}])$. Therefore $\gamma(S, g_h, \mathbb{R}^d) = \min_{s \in S} \gamma(s, g_h, \mathbb{R}^d)$ is also enlarged by the same factor, completing the proof.

Note: Based on the proof, we can see that Theorem 4.2 still holds for ***the case of any data instance***.

## B ADABOOST AND SAMME ALGORITHMS

---
**Algorithm 1** AdaBoost / SAMME
---
1: **Input:** Dataset $\mathcal{D} = \{(x_i, y_i) | 1 \leq i \leq N\}$ size $N$, number of classes $C$ ($y_i \in \{1, 2, ..., C\}, \forall 1 \leq i \leq N$), number of weak learners $M$
2: Initialize sample weight $\boldsymbol{\Lambda} = [\Lambda_i = \frac{1}{N}]_{i=1}^N$
3: **for** $m = 1$ **to** $M$ **do**
4:     Train a classifier $T^{(m)}$ with $\mathcal{D}$ and $\boldsymbol{\Lambda}$
5:     Compute error rate:

$$err^{(m)} = \Sigma_{i=1}^N \Lambda_i \cdot \mathbf{1}_{T^{(m)}(x_i) \neq y_i}$$

6:     Compute the auxiliary weight $\alpha^{(m)}$:
   - **AdaBoost**:

$$\alpha^{(m)} = \log\left(\frac{1 - err^{(m)}}{err^{(m)}}\right)$$

   - **SAMME**:

$$\alpha^{(m)} = \log\left(\frac{1 - err^{(m)}}{err^{(m)}}\right) + \log(C - 1)$$

7:     Update sample weight $\boldsymbol{\Lambda}$:

$$\Lambda_i = \Lambda_i \cdot \exp\{\alpha^{(m)} \cdot \mathbf{1}_{T^{(m)}(x_i) \neq y_i}\}, \forall 1 \leq i \leq N$$

8:     Re-normalize $\boldsymbol{\Lambda}$:

$$\Lambda_i = \frac{\Lambda_i}{\Sigma_{i=1}^N \Lambda_i}, \forall 1 \leq i \leq N$$

9: **end for**
10: **Output:**

$$y(x) = \arg\max_c \sum_{m=1}^M \alpha^{(m)} \cdot \mathbf{1}_{T^{(m)}(x)=c}$$

---

In the realm of classification problems, boosting stands as a formidable strategy. AdaBoost Freund & Schapire (1997) and SAMME Hastie et al. (2009), which addresses Adaboost's limitations in handling multi-class situations, are ensemble learning techniques that combine multiple weak classifiers $\{g^k(\cdot)\}_{k=1}^K$ to create a strong classifier $G(\cdot)$ by sequential training on weighted training data $D = \{(x^i, y^i)\}_{i=1}^N$ including $C$ classes, focusing more on misclassified examples in each iteration $k$ to improve the model performance iteratively as follows:

- For $k = 1$, weak learner $g^k$ is trained on $D$ with the sample weight vector $\boldsymbol{\Lambda} = [\Lambda_i = \frac{1}{N}]_{i=1}^N$. After training, $g^k$ will be evaluated the error rate $err^{(k)} = \sum_{i=1}^N \Lambda_i \cdot \mathbf{1}_{g^k(x^i) \neq y^i}$, and

then the auxiliary weight $\alpha^{(k)} = \log(\frac{1-err^{(k)}}{err^{(k)}}) + \log(C-1)$, which is used to update the sample weight of $D$ for the learning in the next iteration: $\Lambda_i = \Lambda_i \cdot \exp\{\alpha^{(m)} \cdot \mathbf{1}_{g^{(k)}(x^i) \neq y^i}\}, \forall 1 \leq i \leq N$. Following this, the weight of wrongly predicted samples will be increased.

- For $k > 1$, the weak learner $g^k$ will be trained on $D$ with updated $\mathbf{\Lambda}$ from the previous iteration. In which, the sample with a higher corresponding weight will be focused more during the training process. Therefore, the later weak learner will focus more on the mistake of the previous one and will learn to complement that previous classifier. The next operations are similar to the case when $k = 1$.

- Finally, the prediction of $G$ for test sample $x$ is represented by: $y(x) = \arg\max_c \sum_{k=1}^{K} \alpha^{(m)} \cdot \mathbf{1}_{g^{(m)}(x)=c}$.

Detailed pseudocode of both algorithms are provided in Algorithm 1.

Note that applying Adaboost to CL is inherently challenging:

- Adaboost's algorithm requires the whole training set, which is not always available in CL when only accessing the current task's data during training.

- Regarding experimental results, Adaboost often gives superior results with a large enough number of weak learners, usually hundreds or thousands. If there are too few learners, Adaboost will not show a clear advantage (see the results in Table 4).

Fortunately, these challenges have been addressed for the first time in our work, as discussed in section 4.2.2.

## C  PROVING THE THEORY OF OUR MULTI-VIEW RANDOM PROJECTION SCHEME

### C.1  PROOF OF THEOREM 4.3

If we approximate $\boldsymbol{G}$ by $\bar{\boldsymbol{G}}$, the optimal solution is $\boldsymbol{W} = (\bar{\boldsymbol{G}} + \lambda\mathbf{I})^{-1}\tilde{\boldsymbol{Z}}^T\boldsymbol{Y}$. We further have

$$
\bar{\boldsymbol{G}} + \lambda\mathbf{I} = \begin{bmatrix} (\tilde{\boldsymbol{Z}}^1)^T\tilde{\boldsymbol{Z}}^1 + \lambda\mathbf{I} & \mathbf{0} & \mathbf{0} & \mathbf{0} \\ \mathbf{0} & (\tilde{\boldsymbol{Z}}^2)^T\tilde{\boldsymbol{Z}}^2 + \lambda\mathbf{I} & \mathbf{0} & \mathbf{0} \\ \mathbf{0} & \mathbf{0} & ... & \mathbf{0} \\ \mathbf{0} & \mathbf{0} & \mathbf{0} & (\tilde{\boldsymbol{Z}}^k)^T\tilde{\boldsymbol{Z}}^k + \lambda\mathbf{I} \end{bmatrix}
$$

$$
= \begin{bmatrix} \boldsymbol{G}^1 + \lambda\mathbf{I} & \mathbf{0} & \mathbf{0} & \mathbf{0} \\ \mathbf{0} & \boldsymbol{G}^2 + \lambda\mathbf{I} & \mathbf{0} & \mathbf{0} \\ \mathbf{0} & \mathbf{0} & ... & \mathbf{0} \\ \mathbf{0} & \mathbf{0} & \mathbf{0} & \boldsymbol{G}^k + \lambda\mathbf{I} \end{bmatrix}.
$$

$$
\longrightarrow (\bar{\boldsymbol{G}} + \lambda\mathbf{I})^{-1} = \begin{bmatrix} (\boldsymbol{G}^1 + \lambda\mathbf{I})^{-1} & \mathbf{0} & \mathbf{0} & \mathbf{0} \\ \mathbf{0} & (\boldsymbol{G}^2 + \lambda\mathbf{I})^{-1} & \mathbf{0} & \mathbf{0} \\ \mathbf{0} & \mathbf{0} & ... & \mathbf{0} \\ \mathbf{0} & \mathbf{0} & \mathbf{0} & (\boldsymbol{G}^k + \lambda\mathbf{I})^{-1} \end{bmatrix}.
$$

Besides, $\tilde{\boldsymbol{Z}} = [\tilde{\boldsymbol{Z}}^i]_{i=1}^k$ Therefore, we reach

$$
\boldsymbol{W} = \begin{bmatrix} \boldsymbol{W}^1 \\ \boldsymbol{W}^2 \\ ... \\ \boldsymbol{W}^k \end{bmatrix}.
$$

## C.2 PROOF OF COROLLARY 4.4

We derive as follows

$$s^h\left(\boldsymbol{x}, y\right) = \tilde{\boldsymbol{z}}^T \boldsymbol{W} = \left[\begin{array}{cccc} \tilde{\boldsymbol{z}^1}^T & \tilde{\boldsymbol{z}^2}^T & ... & \tilde{\boldsymbol{z}^k}^T \end{array}\right] \left[\begin{array}{c} \boldsymbol{W}^1 \\ \boldsymbol{W}^2 \\ ... \\ \boldsymbol{W}^k \end{array}\right]$$

$$= \sum_{i=1}^{k} s_i\left(\boldsymbol{x}, y\right).$$

## C.3 THE TIGHTNESS OF THE INVERSION APPROXIMATION

We prove that $(\bar{\boldsymbol{G}} + \lambda \mathbf{I})^{-1}$ approximates well $(\boldsymbol{G} + \lambda \mathbf{I})^{-1}$ when $\lambda$ becomes large enough.

Firstly, we show that both $\boldsymbol{G}$ and $\bar{\boldsymbol{G}}$ are positive semi-definite matrices. For any arbitrary vector $\boldsymbol{v} \in \mathbb{R}^{Kd'}$

$$\begin{aligned} \boldsymbol{v}^T \boldsymbol{G} \boldsymbol{v} &= \boldsymbol{v}^T \tilde{\boldsymbol{Z}}^T \tilde{\boldsymbol{Z}} \boldsymbol{v} \\ &= (\tilde{\boldsymbol{Z}} \boldsymbol{v})^T (\tilde{\boldsymbol{Z}} \boldsymbol{v}) \\ &\geq 0 \end{aligned}$$

$$\begin{aligned} \boldsymbol{v}^T \bar{\boldsymbol{G}} \boldsymbol{v} &= \boldsymbol{v}^T \left[\begin{array}{cccc} \tilde{\boldsymbol{Z}^1}^T \tilde{\boldsymbol{Z}^1} & \mathbf{0} & \mathbf{0} & \mathbf{0} \\ \mathbf{0} & \tilde{\boldsymbol{Z}^2}^T \tilde{\boldsymbol{Z}^2} & \mathbf{0} & \mathbf{0} \\ \mathbf{0} & \mathbf{0} & \ldots & \mathbf{0} \\ \mathbf{0} & \mathbf{0} & \mathbf{0} & \tilde{\boldsymbol{Z}^K}^T \tilde{\boldsymbol{Z}^K} \end{array}\right] \boldsymbol{v} \\ &= \boldsymbol{v}^T \left(\left[\begin{array}{cccc} \tilde{\boldsymbol{Z}^1}^T \tilde{\boldsymbol{Z}^1} & \mathbf{0} & \mathbf{0} & \mathbf{0} \\ \mathbf{0} & \mathbf{0} & \mathbf{0} & \mathbf{0} \\ \mathbf{0} & \mathbf{0} & \ldots & \mathbf{0} \\ \mathbf{0} & \mathbf{0} & \mathbf{0} & \mathbf{0} \end{array}\right] + \cdots + \left[\begin{array}{cccc} \mathbf{0} & \mathbf{0} & \mathbf{0} & \mathbf{0} \\ \mathbf{0} & \mathbf{0} & \mathbf{0} & \mathbf{0} \\ \mathbf{0} & \mathbf{0} & \ldots & \mathbf{0} \\ \mathbf{0} & \mathbf{0} & \mathbf{0} & \tilde{\boldsymbol{Z}^K}^T \tilde{\boldsymbol{Z}^K} \end{array}\right]\right) \boldsymbol{v} \\ &= \boldsymbol{v}^T \left[\begin{array}{c} \tilde{\boldsymbol{Z}^1}^T \\ \mathbf{0} \\ ... \\ \mathbf{0} \end{array}\right] \left[\begin{array}{cccc} \tilde{\boldsymbol{Z}^1} & \mathbf{0} & ... & \mathbf{0} \end{array}\right] \boldsymbol{v} + \boldsymbol{v}^T \left[\begin{array}{c} \mathbf{0} \\ \tilde{\boldsymbol{Z}^2}^T \\ ... \\ \mathbf{0} \end{array}\right] \left[\begin{array}{cccc} \mathbf{0} & \tilde{\boldsymbol{Z}^2} & ... & \mathbf{0} \end{array}\right] \boldsymbol{v} + \\ &\quad + \cdots + \boldsymbol{v}^T \left[\begin{array}{c} \mathbf{0} \\ \mathbf{0} \\ ... \\ \tilde{\boldsymbol{Z}^K}^T \end{array}\right] \left[\begin{array}{cccc} \mathbf{0} & \mathbf{0} & ... & \tilde{\boldsymbol{Z}^K} \end{array}\right] \boldsymbol{v} \\ &= \left(\left[\begin{array}{cccc} \tilde{\boldsymbol{Z}^1} & \mathbf{0} & ... & \mathbf{0} \end{array}\right] \boldsymbol{v}\right)^T \left(\left[\begin{array}{cccc} \tilde{\boldsymbol{Z}^1} & \mathbf{0} & ... & \mathbf{0} \end{array}\right] \boldsymbol{v}\right) + \\ &\quad + \left(\left[\begin{array}{cccc} \mathbf{0} & \tilde{\boldsymbol{Z}^2} & ... & \mathbf{0} \end{array}\right] \boldsymbol{v}\right)^T \left(\left[\begin{array}{cccc} \mathbf{0} & \tilde{\boldsymbol{Z}^2} & ... & \mathbf{0} \end{array}\right] \boldsymbol{v}\right) + \\ &\quad + \cdots + \left(\left[\begin{array}{cccc} \mathbf{0} & \mathbf{0} & ... & \tilde{\boldsymbol{Z}^K} \end{array}\right] \boldsymbol{v}\right)^T \left(\left[\begin{array}{cccc} \mathbf{0} & \mathbf{0} & ... & \tilde{\boldsymbol{Z}^K} \end{array}\right] \boldsymbol{v}\right) \\ &\geq 0 \end{aligned}$$

Hence, each eigenvalue of $\boldsymbol{G}$ or $\bar{\boldsymbol{G}}$ is a non-negative real number.

Besides, we also notice that both $\boldsymbol{G} + \lambda \mathbf{I}$ and $\bar{\boldsymbol{G}} + \lambda \mathbf{I}$ are positive semi-definite, which leads to their eigenvalues are also their singular values.

Consider a matrix $\boldsymbol{A}$, denote:

- $rank(\boldsymbol{A})$: rank of $\boldsymbol{A}$
- $\xi_{min}(\boldsymbol{A})$: the smallest eigenvalue of $\boldsymbol{A}$
- $\sigma_{min}(\boldsymbol{A})$: the smallest singular value of $\boldsymbol{A}$
- $\Xi(\boldsymbol{A})$: Set of all eigenvalues of $A$

It is well-known that $\Xi(\boldsymbol{A} + \lambda\mathbf{I}) = \{\xi + \lambda | \xi \in \Xi(\boldsymbol{A})\}$.

From the above we have

$$
\begin{aligned}
\sigma_{min}(\boldsymbol{G} + \lambda\mathbf{I}) &= \xi_{min}(\boldsymbol{G} + \lambda\mathbf{I}) \\
&= \xi_{min}(\boldsymbol{G}) + \lambda \\
&\geq 0 + \lambda \\
&= \lambda
\end{aligned}
$$

And similarly $\sigma_{min}(\bar{\boldsymbol{G}} + \lambda\mathbf{I}) \geq \lambda$.

Secondly

$$
\begin{aligned}
&||(\bar{\boldsymbol{G}} + \lambda\mathbf{I})^{-1} - (\boldsymbol{G} + \lambda\mathbf{I})^{-1}||_F \\
&\leq ||(\bar{\boldsymbol{G}} + \lambda\mathbf{I})^{-1}||_F + ||(\boldsymbol{G} + \lambda\mathbf{I})^{-1}||_F \\
&\leq \sqrt{rank((\bar{\boldsymbol{G}} + \lambda\mathbf{I})^{-1})} \cdot ||(\bar{\boldsymbol{G}} + \lambda\mathbf{I})^{-1}||_2 + \sqrt{rank((\boldsymbol{G} + \lambda\mathbf{I})^{-1})} \cdot ||(\boldsymbol{G} + \lambda\mathbf{I})^{-1}||_2 \\
&\leq \sqrt{Kd'} \cdot (||(\bar{\boldsymbol{G}} + \lambda\mathbf{I})^{-1}||_2 + ||(\boldsymbol{G} + \lambda\mathbf{I})^{-1}||_2) \\
&= \sqrt{Kd'} \cdot \left( \frac{1}{\sigma_{min}(\bar{\boldsymbol{G}} + \lambda\mathbf{I})} + \frac{1}{\sigma_{min}(\boldsymbol{G} + \lambda\mathbf{I})} \right)
\end{aligned}
$$

Assume that $\lambda > 0$, we reach

$$
||(\bar{\boldsymbol{G}} + \lambda\mathbf{I})^{-1} - (\boldsymbol{G} + \lambda\mathbf{I})^{-1}||_F \leq \frac{2\sqrt{Kd'}}{\lambda}
$$

completing the proof.

## D  OUR ALGORITHM FOR DIVERSIFYING MULTIPLE VIEWS WITH BOOSTING PRINCIPLE

This section provides a concise pseudo-code for our proposed technique, which is presented in Section 4.2.2. This step is performed after the backbone is frozen. In general, we need to pre-determine the number of atomic views $K$, and eventually, we obtain $K$ prediction matrices $\mathbf{W}_t^{1:K}$ corresponding to these $K$ views. In particular, for each task $t$, we first initialize the sample weight, a vector whose elements are all 1, corresponding to $n_t$ data samples. Then, we will learn $K$ atomic views in turn; for each view, we calculate the prediction matrix based on (4) and then update the sample weight vector to learn the next view.

---

**Algorithm 2** Multiple-views learning for adapting $T$ tasks sequentially

---

1: **Input:** Data $\{\mathcal{D}_t = (\boldsymbol{X}_t, \boldsymbol{Y}_t)\}_{t=1,\dots T}$, Number of atomic views: $K$
2: **for** task $t = 1$ **to** $T$ **do**
3:     Initialize observation weights for data samples of $\mathcal{D}_t$:
        $\boldsymbol{\Lambda}_t^0 = [\Lambda_{t,i}^0 = 1]_{i=1}^{n_t}$
4:     **for** view $k = 1$ **to** $K$ **do**
5:         Calculate $\mathbf{W}_t^k$ of $k^{th}$ view incrementally using (4) with weights $\boldsymbol{\Lambda}_t^{k-1}$.
6:         Update the weights $\boldsymbol{\Lambda}_t^k$ according to (2).
7:     **end for**
8: **end for**
9: **Output:** Prediction matrices $\mathbf{W}_t^{1:K}$

---

Looking closer at the algorithm, since the prediction matrices are calculated based on accumulating knowledge from data of tasks so far, thus we used $\boldsymbol{\Lambda}_t^0 = [\Lambda_{t,i}^0 = 1]_{i=1}^{n_t}$ for initialization (Line 3), instead of $\boldsymbol{\Lambda}_t^0 = [\Lambda_{t,i}^0 = \frac{1}{n_t}]_{i=1}^{n_t}$ as in the original Algorithm 1. Besides, to create the balance when learning common classifiers for the sequence of tasks, we also perform a normalization step for this vector before forming $\boldsymbol{\Lambda}^k$ for $k^{th}$ view.

## E PROMPT TUNNING FOR TASK-SPECIFIC ADAPTATION

Figure 3 below illustrates more clearly the motivation of our method in utilizing a task-wise prompt-based task adaptive backbone.

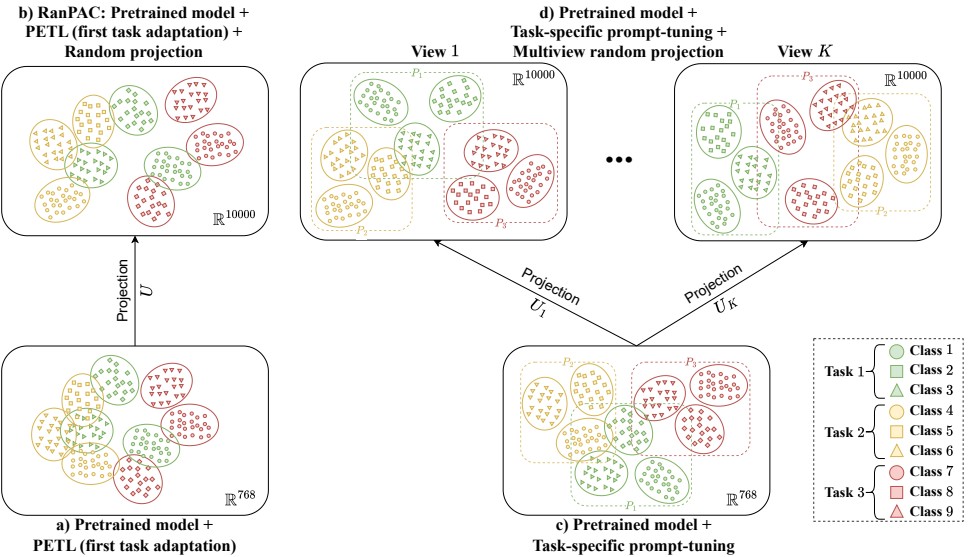

Figure 3: (a)&(b) Visualization of potential overlapping in later tasks of RanPAC in original latent space and high dimensional space due to the lack of adaptation to these tasks. (c) With task-wise prompting, the original features of the classes tend to be better distinguished. (d) Multi-view random projection encourages clearer separations in different views.

Inspired by HiDE, we maintain a set of base prompts $\boldsymbol{Q}_{1:t}$ up to task $t$ and compute the prompt $\boldsymbol{P}_t$ for task $t$ as $\boldsymbol{P}_t = (1 - \zeta)\boldsymbol{Q}_t + \zeta \sum_{i=1}^{t-1} \boldsymbol{Q}_i$, where $\zeta$ is a parameter within the range of $[0, 1]$. During learning task $t$, the base prompts $\boldsymbol{Q}_{1:t-1}$ are fixed. After that, we insert $\boldsymbol{P}_t$ into the pre-trained backbone and train the current task using Cross-Entropy loss and a contrastive-based regularization loss as follows:

$$\min_{\theta, \boldsymbol{P}_t} CE\left(\boldsymbol{P}_t, \theta\right) + \beta Reg\left(\boldsymbol{P}_t\right) \tag{30}$$

where $\beta > 0$ is a trade-off parameter and $\theta$ is the weights of temporary classification head of prompted model $f_{\Phi, \boldsymbol{P}}$.

In particular, to learn representation within the current task, we use Cross Entropy loss:

$$CE(\boldsymbol{P}_t, \theta) = \frac{1}{n_t} \sum_{i=1}^{n_t} CE(h_\theta(f_{\Phi, \boldsymbol{P}_t}(\boldsymbol{x}_t^i)), y_t^i), \tag{31}$$

where $h_\theta$ is the auxiliary classification head we employ to learn to representation $\boldsymbol{z}_t = f_{\Phi, \boldsymbol{P}_t}(\boldsymbol{x})$.

In addition, to enforce the separation between features of new task $t$ and previous ones, at the end of each previous task, we summarize and store the representations of a class $k$ in an old task to a mean

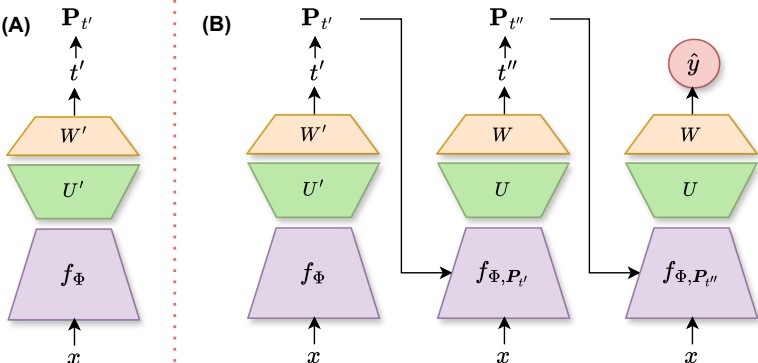

Figure 4: Prompt selection process (A) and Self-improvement prompt selection strategy (B)

vector $\boldsymbol{m}_k$ by simply averaging those class representations and use the following contrastive loss:

$$Reg(\boldsymbol{P}_t) = \frac{1}{n_t} \sum_{i=1}^{n_t} \frac{1}{|\mathcal{Y}_{t-1}|} \sum_{k=1}^{|\mathcal{Y}_{t-1}|} \ell(\boldsymbol{z}_t^i), \tag{32}$$

where $\ell(\boldsymbol{z}_t^i) = \dfrac{\exp\left(\boldsymbol{z}_t^i.\boldsymbol{m}_k/\tau\right)}{\sum_{j=1}^{n_t} \exp\left(\boldsymbol{z}_t^i.\boldsymbol{z}_t^j/\tau\right) + \sum_{k=1}^{|\mathcal{Y}_{t-1}|} \exp\left(\boldsymbol{z}_t^i.\boldsymbol{m}_k/\tau\right)}$ and $\tau$ is set to 0.8.

Besides, Figure 4 illustrates more clearly our prompt selection process and our proposed self-improvement strategy, which are described in detail in Section 4.3.

## F EXPERIMENTAL SETTINGS

### F.1 DATASETS

We follow the protocols in (McDonnell et al., 2023) to construct the following common benchmarks:

- **Split CIFAR-100** Krizhevsky et al. (2009): This dataset comprises images from 100 classes, each of a small scale. These classes are divided randomly into 10 separate incremental tasks, with each task featuring a distinct set of classes.

- **Split ImageNet-R** Krizhevsky et al. (2009): This dataset comprises images from 200 classes, categorized as large-scale. It includes challenging examples from the original **ImageNet** dataset and newly gathered images representing diverse styles. These classes are also randomly divided into 10 distinct incremental tasks.

- **5-Datasets** Ebrahimi et al. (2020): This composite dataset incorporates **CIFAR-10** Krizhevsky et al. (2009), **MNIST** LeCun et al. (1998), **Fashion-MNIST** Xiao et al. (2017), **SVHN** Netzer et al. (2011), and **notMNIST** Bulatov (2011). Each of these datasets is treated as a separate incremental task. This structure allows for the assessment of the effects of significant variations between tasks.

- **Split CUB-200** Wah et al. (2011): This dataset consists of fine-grained images of 200 different bird species. Like the others, it is randomly segregated into 10 incremental tasks, each comprising a unique class subset. Note that we use the same number of training samples as in RanPAC (McDonnell et al., 2023), which is larger than that used in SLCA (Zhang et al., 2023).

### F.2 BASELINES

In the main paper, we use CL methods with pre-trained ViT as the backbone. We group them into CP-based and Prompt-based approaches.

(1) **RanPAC** (McDonnell et al., 2023): a CP-based method that adapts the first task using parameter-efficient transfer learning, then projects the features onto a high dimensional space, and finally incrementally learns a linear classifier. Different from our method, RanPAC only performs adaptation at the first task and uses one projection matrix, resulting in potential overlapping of future tasks' features and infeasibility when working with extremely large dimensions.

(2) **ADaM**(Zhou et al., 2023): another CP-based work that proposes a framework which adapts the pre-trained model to the first task using several adapting techniques such as prompt-tuning (Jia et al., 2022), adaptor (Chen et al., 2022) and batch-norm statistics adjustment, so that the representation could be more specific towards the downstream CL task. At prediction, ADaM only uses the nearest class mean classifier, which is shown in (Goswami et al., 2023) to be sub-optimal as it lacks second-order statistics such as the covariance matrix.

(3) **SLCA**: (Zhang et al., 2023): SLCA is a novel method that proposes learning new tasks with a small learning rate for the feature extractor and a slightly larger one for the classification head so that the model can reduce the potential representation shift. It additionally implements a classifier alignment based on saved features to improve the classification head at inference further. On the one hand, in SLCA, using a small learning rate will lead to less-shifted representation. However, this also makes the model become too stable to learn new tasks. This limits the performance of SLCA.

(4) **L2P** Wang et al. (2022d): the first prompt-based work for CL that proposed using a common prompt poll, in which the top $k$ most suitable prompts are selected for each sample for training and testing. This might allow knowledge transfer between tasks but also carries the potential risk of catastrophic forgetting. In addition, different from us, L2P has not really considered training classifiers as well as setting constraints on features from old and new tasks during training, which likely leads to limitations in the model's predictability.

(5) **DualPrompt** Wang et al. (2022c) is the prompt-based method that attempts to overcome the disadvantages of L2P when attaching complementary prompts to the pre-trained backbone instead of only prompting at input. Besides, DualP also integrates additional prompt sets for each task to exploit more task-specific "instruction" in addition to task invariant information from the common pool. However, like L2P, this method does not really focus on learning classification head efficiently. Moreover, choosing the wrong promptID (for task-specific instruction) during testing could also reduce model performance.

(6) **S-Prompt++** Wang et al. (2022b): is originally proposed for Domain-incremental learning. It trains a separate prompt and classifier head for each task. At evaluation, it infers domain id as the nearest centroid obtained by applying K-Means on the training data. To adapt S-Prompt to CIL, S-Prompt++ uses one common classifier head for all tasks. This method also has the same limitations as pointed out in DualP: the learning of classification head and the prediction of appropriate prompts at testing time.

(7) **CODA-Prompt** Smith et al. (2023): This prompt-based work uses task-specific learnable prompts for each task, but similar to L2P, CODA employs a pool of prompts and keys and then computes a weighted sum from these prompts to generate the real prompt. The weights are defined as the cosine similarity between queries and keys. At the end of the task sequence, to avoid task prediction, its weighted sum always considers all the prompts. CODA offers improvements over DualP and L2P in that the optimization of keys and prompts occurs simultaneously, but it still has not solved the drawbacks we mentioned about DualP above.

(8) **HiDe-Prompt** Wang et al. (2023): a recent SOTA prompt-based method that decomposes learning CIL into 3 modules: a task inference, a within-task predictor and a task-adaptive predictor. The second module trains prompts for each task with a contrastive regularization that tries to push features of new tasks away from prototypes of old ones. To predict task identity, it trains a classification head on top of the pre-trained ViT. TAP is similar to a fine-tuning step that aims to alleviate classifier bias using the Gaussian distribution of all classes seen so far. Although HiDE solves one of the limitations of the above methods, in comparison with ours, this method still suffers from classifier forgetting, and its taskID predictor may be sub-optimal if the backbone is too different from downstream CL tasks, which we tackle by our self-improvement process.

(9) **CPP** Li et al. (2024): This recent SOTA also uses a contrastive constraint to control features of all tasks so far during representation learning and achieves roughly equivalent performance to HiDE on

the same settings. Nevertheless, this method still has the advantages that we pointed out in HiDE, which we propose to address in our work.

### F.3 METRICS

In our study, we employed two key metrics: the Final Average Accuracy (FAA) and the Final Forgetting Measure (FFM). To define these, we first consider the accuracy on the $i$-th task after the model has been trained up to the $t$-th task, denoted as $A_{i,t}$. The average accuracy of all tasks observed up to the $t$-th task is calculated as $AA_t = \frac{1}{t} \sum_{i=1}^{t} A_{i,t}$. Upon the completion of all $T$ tasks, we report the Final Average Accuracy as FAA $= AA_T$. Additionally, we calculate the Final Forgetting Measure, defined as FFM $= \frac{1}{T-1} \sum_{i=1}^{T-1} \max_{t \in \{1,...,T-1\}} (A_{i,t} - A_{i,T})$. The FAA serves as the principal indicator for assessing the ultimate performance in continual learning models, while the FFM evaluates the extent of catastrophic forgetting experienced by the model.

### F.4 IMPLEMENTATION DETAILS

In our method, we set the contrastive regularization weight $\beta = 0.1$, the parameter for prompt construction $\zeta = 0.1$, and the confidence threshold for expert filtering is $0.5$. The default values for the number of self-improvement steps is 2, the number of expert views is 15, and the dimension of each atomic view is 10,000.

Our implementation aligns with the methodologies employed in prior research Wang et al. (2022d;c); Smith et al. (2023). Specifically, our framework incorporates the use of a pre-trained Vision Transformer (ViT-B/16) as the backbone architecture. For the optimization process, we utilized the Adam optimizer, configured with hyper-parameters $\beta_1$ set to $0.9$ and $\beta_2$ set to $0.999$. The training process was conducted using batches of 128 samples, and a fixed learning rate of $0.005$ was applied across all models except for CODA-Prompt. For CODA-Prompt, we employed a cosine decaying learning rate strategy, starting at $0.001$. Additionally, a grid search technique was implemented to determine the most appropriate number of epochs for effective training. Regarding the pre-processing of input data, images were resized to a standard dimension of $224 \times 224$ pixels and normalized within a range of $[0, 1]$ to ensure consistency in input data format.

In Table 1 of the main paper, the results of L2P, DualPrompt, S-Prompt++, CODA-Prompt, and HiDe-Prompt on Split CIFAR-100 and Split ImageNet-R are taken from (Wang et al., 2023). Their results on the other two datasets are produced from the official code provided by the authors. For CPP, the reported results are also reproduced from their official code. Besides, the results of ADaM and RanPAC on Split CIFAR-100, Split ImangeNet-R, and CUB are also taken directly from RanPAC paper (McDonnell et al., 2023), while the remaining is produced by their official code.

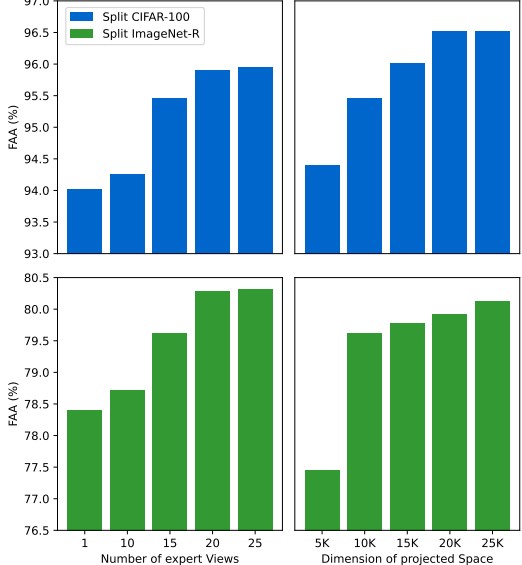

Figure 5: Performance when the number of atomic views and the dimension of each projected space (of each atomic view) vary.

### G ADDITIONAL EXPERIMENTS

#### G.1 EXPERIMENTS
ABOUT OUR ENSEMBLE CLASSIFIER

**Effect of the number of atomic views and their dimension.** Figure 5 illustrates that overall, our model's FAA gradually improves when increasing the values of these two quantities. Notably, a breakthrough occurs when increasing from 10 to 15 views on Split CIFAR-100 and from 15 to 20 views on Split ImageNet-R. In addition, on both datasets, the most significant improvement is observed when changing the dimension of the

projected space from 5,000 to 10,000, with an increase of more than 2%. Besides, we can observe the convergences of FAA as the two quantities reach certain values. This might be due to the limitation of the linear classifiers.

**The effectiveness of our voting strategy.** Table 4 presents additional experimental results showcasing the advantages of our proposed voting strategy. Our method with voting strategy **(B)** outperforms the single-view version (**D**, or BoostCL-s), multi-view without boosting version (**C**, whose classifiers on different atomic views are trained independently), and AdaBoost **(A)** on two considered datasets. In particular, using AdaBoost (A) only brings marginal or even negligible improvement compared to using the single view model (D). This may be due to the limited number of views/learners ($K = 15$), which is insufficient for AdaBoost, typically requiring numerous iterations/weak learners to demonstrate its strength. Meanwhile, with the given number of atomic views, our voting method can generate at most $2^{15} - 1$ responses, leading to better efficiency. Furthermore, simply using multi-view without boosting will not be effective, as it only yields performance equivalent to the single-view version. This is because the output of the respective views is almost the same in most cases.

Table 4: Comparison between our voting strategy and AdaBoost

| Method | Benchmark | |
| --- | --- | --- |
| | Split CIFAR-100 | Split ImageNet-R |
| (A) AdaBoost (15 leaners) (Boosting w/o voting) | 94.15 | 78.55 |
| (B) Multi-view ($K = 15$) (Boosting & voting) | **95.45** | **79.62** |
| (C) Multi-view ($K = 15$) (w/o boosting) | 94.03 | 78.52 |
| (D) Singe view | 94.03 | 78.52 |

**Applicability of our ensemble classifier.** Table 5 shows that our ensemble classifier can flexibly adapt to other prompt-based and CP-based methods and enhance their performance. We observe that as long as $f_{\Phi, P_{t'}}(x)$ does not shift much from $f_{\Phi, P_t}(x)$, the linear classifier with the weight matrix $W$ on the main label prediction branch can sufficiently generalize to handle this possible shift. The results depict the efficiency of our voting ensemble classifier in enhancing the performance of baselines, where RanPAC ++, SLCA++, and HIDE++ are the versions of RanPAC, SLCA, and HIDE when using our ensemble classifier, respectively, after training the respective backbone.

Table 5: Using our ensemble classification head for baselines

| Method | Benchmark | |
| --- | --- | --- |
| | Split CIFAR-100 | Split ImageNet-R |
| RanPAC | 92.20 | 77.90 |
| RanPAC ++ | 93.62 | 78.62 |
| SLCA | 91.62 | 77.10 |
| SLCA ++ | 92.82 | 77.56 |
| HiDE | 92.61 | 75.06 |
| HIDE ++ | 93.82 | 76.01 |
| BoostCL-s | 94.03 | 78.52 |
| BoostCL-m | **95.45** | **79.25** |

**In the absence of Random Projection**. To further demonstrate the benefit of out ensemble classifier, we remove the random projection, i.e. applying the multi-view strategy directly on the original latent space. The results of this method are presented in Table 6. It can be seen that without Random Projection, hence no additional parameters for the projection matrix, our BoostCL still significantly outperforms RanPAC and HiDE. Here we also include BoostCL-m (d' = 768; K = 13) which has roughly the same number of parameter compared to RanPAC. This version's performance is slighly lower than the one without RP, showing that the benefit of random projection tends to occur when $d'$ is sufficiently large (c.f, in Figure 5).

Table 6: Performance (FAA - %) of our method with and w/o using RP.

| Method | Benchmark | |
| --- | --- | --- |
| | Split CIFAR-100 | Split ImageNet-R |
| RanPAC | 92.20 | 77.90 |
| HiDE | 92.61 | 75.06 |
| BoostCL-s | 94.03 | 78.52 |
| BoostCL-m (d' = 768; K = 13) | 94.56 | 78.65 |
| BoostCL-m (d' = 10, 000; K = 15) | **95.45** | **79.25** |
| BoostCL-m w/o RP & K = 13 | 94.65 | 78.82 |

## G.2 DETAILED ABLATION STUDY

To summarize each of our proposed components' pros, we provide detailed ablation study in Table 7 with the following remarks:

- Line 6 > RanPAC: the merits of Task-adaptive backbone.
- Line 4 = Line 5 = Line 6: multi-view without boosting is useless as all the views produce the same prediction.
- Line 3 > Line 5: AdaBoost only brings slight improvement.
- Line 2 > Line 3: Our voting strategy outperforms AdaBoost.
- Line 7 > Line 6 and Line 1 > Line 2: Self-improvement process is helpful.

(Symbols $>, =$ refers to the comparison between two values of FAA).

Table 7: More detailed ablation study on Split-CIFAR-100

| ID | self-improve | voting | boosting | RP(d'=10k, K=15) | RP(d'=10k, K=1) | FAA (%) |
| --- | --- | --- | --- | --- | --- | --- |
| 1 | ✓ | ✓ | ✓ | ✓ | × | **96.55** |
| 2 | × | ✓ | ✓ | ✓ | × | 95.45 |
| 3 | × | × | ✓ | ✓ | × | 94.15 |
| 4 | × | ✓ | × | ✓ | × | 94.03 |
| 5 | × | × | × | ✓ | × | 94.03 |
| 6 | × | × | × | × | ✓ | 94.03 |
| 7 | ✓ | × | × | × | ✓ | 94.72 |

## G.3 COMPLEXITY ANALYSIS (COMPUTATION AND STORAGE COSTS)

In this section, we present a detailed analysis of our model's computational costs compared to baselines. Then, we show details related to our ensemble classifier and the self-improvement process. The reported results were conducted on 2 machines with Tesla V100-SXM2-32GB-LS.

### G.3.1 TIME COMPLEXITY OF MODELS

Table 8 provides the time complexity of our method (w/o self-improvement process) and the two strongest baselines (RanPAC and HiDE), in terms of forward time and training time.

Table 8: Computational time, measured on Split CIFAR-100 (each image has size $3 \times 224 \times 224$).

| Method | Forward time (ms/sample) | Training time (min/task) |
| --- | --- | --- |
| RanPAC | $12.21 \pm 1.04$ | $1.42 \pm 0.22$ |
| HiDE | $21.88 \pm 1.08$ | $42.62 \pm 1.35$ |
| BoostCL | $27.80 \pm 1.22$ | $65.52 \pm 1.60$ |
| BoostCL (backbone) | $24.02 \pm 1.12$ | $44.17 \pm 1.52$ |
| BoostCL (ensemble classifier) | $3.89 \pm 0.31$ | $21.41 \pm 1.15$ |

**Forward time:**

- (1) In general, BoostCL takes the largest amount of time, due to exploiting both prompt-based backbone and ensemble classifier. However, the cost gap between our method and the strongest prompt-based baseline, HiDE, is insignificant.

- (2) The most expensive process is for the backbone forwarding. Meanwhile, the time spent on our proposed ensemble classifier, which works on a set of high-dimensional linear classifiers, only accounts for a small fraction ($14\%$) compared to the total forward time.

**Training time:**

We can see that the time spent training our ensemble classifier ($d' = 10,000; K = 15$) accounts for nearly one-third of the total time for each task. The reason is that our approach exploits the boosting strategy so that we have to train atomic views sequentially given that the time for training each view is mostly equal to the time for training each task of RanPAC ($d' = 10,000$). However, we believe this cost can be offset by the improvement we got.

### G.3.2 TIME/SPACE COMPLEXITY OF OUR ENSEMBLE CLASSIFIER

To further analyze the cost and efficacy obtained when using random projection and multiple-view, we compare our ensemble classifier and RanPAC's classifier (Table 2):

**Storage cost:**

- When training BoostCL ($d' = 10,000; K = 15$), we exclusively use one Random Projection matrix for all atomic views. We also store internally, e.g, put to GPU, the Gram matrix and weight vector corresponding to the atomic view that is being trained, while the Gram matrices and weight vectors of other views are stored externally. Therefore, essentially, the internal memory utilized for training our ensemble classifier (BoostCL with $d' = 10,000; K = 15$) is equivalent to that of RanPAC ($d' = 10,000$). Thus, in the table below, we only report the number of trained parameters used at testing time.

- When testing, we can see that the classifier of the version of BoostCL ($d' = 10,000; K = 15$) requires 15 times more memory than RanPAC ($d' = 10,000$). However, this expense can be compensated by the more-than-3% improvement in FAA that our BoostCL obtains. Moreover, considering BoostCL ($d' = 768; K = 13$), which requires less memory than RanPAC, we can see that with the help of our ensemble strategy, it can still improve the FAA of RanPAC and even BoostCL single view ($d' = 10,000$).

- For the case that RanPAC uses $d' = 10,000 \times 15 = 150,000$, the total dimension in BoostCL. The cost to store the Gram matrix (with size $d' \times d'$) or its inversion is too huge and impractical. Meanwhile, BoostCL works well with this setting, as it only needs to deal with submatrices.

**Regarding the computational complexity:**

- Comparing with the original RanPAC ($d' = 10,000$), forwarding the classifier of BoostCL ($d' = 10,000; K = 15$) requires more time. This is obvious because we use an ensemble strategy for prediction instead of using only one classification head like RanPAC. This cost is offset by the performance improvement we achieve.

- Last but not least, when discussing the potential of using projection matrices on high-dimensional space, we claimed in our paper that our process helps to alleviate the "substantial computational cost when computing the inverse of a $d' \times d'$ Gram matrix for solving the linear classifier" which is done in RanPAC. In this table, we compare our BoostCL with RanPAC$*$ ($d' = 10,000 \times 15$) and mark that this version of RanPAC is infeasible to train because the cost when computing the inversion of a Gram matrix of size $d' (= 10,000 \times 15)$ is $O(d'^c)$ for some constant $c \in [2.3, 3.0]$, according to Geéron (2017). Therefore, BoostCL has solved this computational crisis, saving the time cost for a factor of $O(15^c)$.

### G.3.3 TIME CONSUMPTION OF OUR SELF-IMPROVEMENT STRATEGY

We provide the experimental results depicting the time consumption of our self-improvement strategy in Table 9. According to the results, such each step costs about 12ms for a sample. However, we just need 2 prompt prediction steps to significantly improve performance. This is worth compensating.

Table 9: Performance when increasing the number of prompt prediction steps (Split CIFAR-100).

| Metric | Number of steps | | | | |
|--------|------|------|------|------|------|
| | 1 | 2 | 3 | 4 | 5 |
| FAA (%) | 95.45 | 96.55 | 96.85 | 97.03 | 97.05 |
| Time (ms) | 27.80 | 39.02 | 50.95 | 61.84 | 73.85 |

### G.4 INVERTIBLE AND EXPANSIVE NON-LINEAR ACTIVATION FUNCTIONS FOLLOWING THE RP

The Table 10 below depicts model performance across different datasets when using the ReLU, Leaky ReLU (negative slop $\beta = 0.01$), and the activation function $a(x) = x + \xi \cdot \log(1 + \exp(x))$ where $\xi = 100$, which we used in our main experiments. The results show that the obtained FAA corresponding to these activation functions do not differ too much. This may be because these three functions are basically nearly equivalent, plus the property of the datasets make the results not differ too much, making ReLU still work well when combined with RP although it violates the invertible condition in the region of $x < 0$. We leave a detailed explanation for future research.

Table 10: Performance when using different non-linear activation functions.

| Activation function | Split CIFAR-100 | | Split Imagenet-R | |
|---------------------|-----------|--------|-----------|--------|
| | BoostCL-s | RanPAC | BoostCL-s | RanPAC |
| Leaky ReLU | 93.92 | 92.12 | 78.43 | 77.88 |
| ReLU | 93.93 | 92.20 | 78.45 | 77.90 |
| Our $a(x)$ | 94.03 | 92.25 | 78.52 | 77.96 |

