# OpenReview forum: "Boosting Multiple Views for pretrained-based Continual Learning"
_ICLR.cc/2025/Conference — ICLR 2025 Poster_

### Official Review · Reviewer_H1Kx · 2024-10-28

**Soundness:** 3
**Presentation:** 3
**Contribution:** 3
**Rating:** 6
**Confidence:** 4

**Summary:**

This paper provides some theoretical analysis on Random Projection (RP) which project features into a higher-dimensional space to improve their linear separability. Furthermore, this paper proposes a Multi-View Random Projection scheme to create a robust ensemble classifier motivated by AdaBoost. Then the proposed method is applied to prompt-based CL methods for a better performance. Experimental results demonstrate its efficacy in various continual learning settings.

**Strengths:**

1.	This paper provides theoretical insights into Random Projection (RP), reinforcing its foundation in an intuitive manner.

2.	The motivation for the proposed Multi-View Random Projection is clear and well-justified, demonstrating its potential to enhance performance.

3.	The paper is well-organized and easy to understand, addressing most of my queries within the main text or the appendix.

**Weaknesses:**

**Major**

1.	The necessity of the huge views should be clarified. As mentioned in the paper, the subsequent atomic views aim to capture the complement knowledge of previous views. However, the final classifier only leverages a part of these views rather than all of them, which may omit some complement knowledge.

2.	The theoretical results are interesting, but the method is heuristic. For example, the design of the voting strategy and the self-improvement are quite complex. It seems that a lot of tricks are introduced directly to improve performance.

3.	The design of the self-improvement process is not intuitive, which may accumulate errors in multiple-step prediction. Specifically, if an incorrect prompt is chosen in the first step, the model may become biased towards predicting the class associated with that wrong prompt, potentially leading to accumulated errors and hindering improvements in prompt prediction accuracy.

4.	The experiments do not include comparisons with simpler ensemble methods, such as randomly sampling views or using all views for the ensemble. Such comparisons can provide more intuitive insights to demonstrate the effectiveness of the proposed method.

**Minor**

1.	There are too many notations which may limit the fluency and readability. For example, in line 177, there is no description for the notation $\gamma (s,g,\mathbb{R}^d)$ and no explanation for the symbolic $g$. So please simplify the notations as much as possible.
2.	Typo in Corollary 4.4 (line 250), “sammple” should be “sample”.

**Questions:**

Please refer to weaknesses.

---

> ### Author Response · Authors · 2024-11-22
> **Response to Reviewer H1Kx**
>
> We thank the reviewer for taking time to review and appreciating our paper. We would like to respond to the main questions and concerns as follows:
>
> **1. Necessity of the huge views should be clarified. As mentioned in the paper, the subsequent atomic views aim to capture the complement knowledge of previous views. However, the final classifier only leverages a part of these views rather than all of them, which may omit some complement knowledge.**
>
> - We believe that **the necessity of a huge view with very high dimension $d'$ is well-motivated in our paper from both theoretical and practical perspectives**. Specifically, in **Section 4.1**, we theoretically prove that a high dimensional projection space leads to higher margins of classifiers, hence boosting the generalization ability. Moreover, in **Appendix G.1 and Figure 5**, we experimentally verify this theoretical finding.
> - Additionally, in **Section 4.2.1**, we explicitly discuss that although a very high projection dimension $d'$ is necessary, its induced computation is prohibitive. Hence, it motivates us to come up with learning a sequence of $K$ atomic views of dimensions $d_a$ with $d' = Kd_a$. However, we argue that learning independent classifier heads for these atomic views does not improve the performance because the outputs are less diverge.
> - Therefore, we leverage with AdaBoost principle to learn complementary classifier heads for the atomic views. It is worth noting that tailoring AdaBoost for CL in the context of prompt-based learning and random projection is an essential contribution of our paper. **The main aim of creating complementary views, as inspired by Adaboost, is creating a diverse pool of atomic views**. By trying to complement the previous view, each of these views tries to maximize the performance of the model from a different perspective.
> - More importantly, AdaBoost always requires many component classifiers. To this end, we combine some atomic views to gain a huge view. Certainly, at first glance, one needs to train a classifier head for a huge view, which seems to be expensive due to the number of $2^K -1$ huge views. Fortunately, in our **Theorem 4.3 and Corollary 4.4**, we demonstrate the classifier on the huge view can be computed based on the ones on the atomic views. Therefore, we can conveniently create at most $2^K -1$ classifiers for huge views for boosting.
> - Regarding our voting strategy, we do not use all $2^K -1$ outputs for voting. Specifically, we first select the atomic views with sufficient confidence levels, denoted as $A$. We then combine atomic views in $A$ to form $2^{|A|} -1$ huge views for voting. The reason is that **the atomic views that are less confident seem to provide uncertain predictions, hence less sharply contributing to the summative final decision, or even harm the performance, as reported in the table below. Moreover, by eliminating some less confident atomic views, we can reduce the inference time.** Please note that **threshold =  0** means that we choose all atomic views for creating $2^K -1$ huge views for voting.
>
>   | Threshold | 0|0.2 | 0.5 | 0.6 | 0.7 | 0.9 |
>   | --- | --- | --- | ---|---|---|---|
>   | Split-CIFAR-100|  93.22 | 94.65 | 95.45  | 95.03 | 94.21 | 94.03|
>   | Split-Imagenet-R| 78.03 | 78.98 | 79.62  | 79.22  | 78.52|# |
>
>
>
> **2. The theoretical results are interesting, but the method is heuristic. For example, the design of the voting strategy and the self-improvement are quite complex. It seems that a lot of tricks are introduced directly to improve performance.**
>
> - **Our proposed approach has its root in the well-established and well-grounded AdaBoosting**. Our way to create many huge views for voting is innovative and well-motivated. Moreover, our approach of using the confidence level to restrict the number of classifiers on huge views for final voting is also well-motivated and intuitive.
>
> - Regarding the **self-improvement process, this is our minor contribution—simple but efficient**. Specifically, the more iterations used for choosing the prompt ID, the more likely the correct prompt ID could be selected, resulting in higher accuracy. We believe the effectiveness of this technique will provide valuable insights for future work. Additionally, **even without this self-improvement, our approach significantly outperformed the state-of-the-art baselines.**
>
> - In our method implementation, all terms are implemented following the well-motivated components as described in the paper.

---

> ### Author Response · Authors · 2024-11-23
> **Response to Reviewer H1Kx (2)**
>
> **3. The design of the self-improvement process is not intuitive, which may accumulate errors in multiple-step prediction. Specifically, if an incorrect prompt is chosen in the first step, the model may become biased towards predicting the class associated with that wrong prompt, potentially leading to accumulated errors and hindering improvements in prompt prediction accuracy.**
>
> - Practically, we observe that the performance of the final prediction (i.e., predicting labels of data with prompts integrated) is always higher than the promptID prediction. This observation motivated our proposed technique. Although relatively simple, it is elegant and could offer valuable insight for future work.
>
> - Sometimes, the wrong prediction is inevitable, as the reviewer has concerned. However, as mentioned in our paper (**Lines 395-399**), even when we pick a wrong prompt, we still get the correct final answer.
>
> **4. The experiments do not include comparisons with simpler ensemble methods, such as randomly sampling views or using all views for the ensemble. Such comparisons can provide more intuitive insights to demonstrate the effectiveness of the proposed method.**
>
> Please refer to **Table 4 - Appendix G.1** (or below)
> In this table, we compare the performance of our method with that of other simpler ensemble methods.
>
> | Method                                     | Split CIFAR-100 | Split ImageNet-R |
> |------------|------------------|-------------------|
> | (A) AdaBoost (15 learners) (Boosting w/o voting) | 94.15            | 78.55             |
> | (B) Multi-view (K = 15) (Boosting & voting)      | **95.45**            | **79.62**             |
> | (C) Multi-view (K = 15) (w/o boosting)           | 94.03            | 78.52             |
> | (D) Single view                               | 94.03            | 78.52             |
>
> Particularly, the results of using all views are fully reported in the case of boosting strategy (A) and when trained naively (C). Randomly sampling views without our strategy would give the same results as (C) since the views give almost identical results.
>
> **5. Regarding notations and typos**
>
> - The symbolic $g$ denotes the classifier, which we mentioned before in **Line 181.**
>
> - We have fixed the typos accordingly.

---

> > ### Comment · Reviewer_H1Kx · 2024-11-28
> > **Thanks for the response**
> >
> > Thanks for the authors' detailed and clear response. Most of my concerns have been addressed and I will keep my score.

---

> ### Author Response · Authors · 2024-11-28
>
> Dear Reviewer H1Kx,
>
> Thank you for your response. We are glad that you feel we have adequately addressed most of your concerns.
>
> Could you please let us know what we should do to further address your concerns?
> Your detailed feedback will definitely help enhance the quality of our work.

---

### Official Review · Reviewer_gjF6 · 2024-10-31

**Soundness:** 3
**Presentation:** 3
**Contribution:** 2
**Rating:** 5
**Confidence:** 5

**Summary:**

This manuscript introduces "BoostCL", a novel approach to improve continual learning based on pre-trained models. The manuscript provides theoretical analysis to show that feature separability and model generalization ability benefit from random projection. Then a multi-view random projection is proposed to adapt AdaBoost for CL. A self-improvement process is also proposed for appropriately selecting prompts for each task sample. Empirical validation on multiple CL benchmarks is then conducted.

**Strengths:**

1. The manuscript provides theoretical analysis of Random Projection, demonstrating how RP improves feature separability in high-dimensional space.
2. The manuscript is clear and well organized. The formulas are presented in an unambiguous manner.

**Weaknesses:**

A substantive assessment of the weaknesses of the paper. Focus on constructive and actionable insights on how the work could improve towards its stated goals. Be specific, avoid generic remarks. For example, if you believe the contribution lacks novelty, provide references and an explanation as evidence; if you believe experiments are insufficient, explain why and exactly what is missing, etc.

1. Although RP and multi-view strategies are theoretically shown to improve feature separability, there is limited discussion on the theoretical upper bound of generalization error.
2. The manuscript does not provide theoretical guidance on choosing the optimal projection dimension in RP, which can greatly impact model performance for different tasks.
3. The proposed self-improvement process for prompt selection could increase the model’s inference complexity, but this potential trade-off is not fully analyzed, especially in scenarios with many tasks.
4. Minor spelling errors, like “sammple” in line 250.
5. More experiment regarding hyper-parameters like number of views, projection dimensions, and threshold settings for the voting mechanism should be conducted.

**Questions:**

1. How to effectively choose the best projection dimension to make appropriate trade-off between performance and computational overhead.
2. How much gpu resource does the proposed method need, will the gpu memory usage increase rapidly by increasing projection dimension.

---

> ### Author Response · Authors · 2024-11-22
> **Response to Reviewer gjF6**
>
> We thank the reviewer for your solid and constructive comment. We would like to respond to your concerns as follows:
>
> **2. The manuscript does not provide theoretical guidance on choosing the optimal projection dimension in RP, which can greatly impact model performance for different task. How to effectively choose the best projection dimension to make appropriate trade-off between performance and computational overhead.**
>
> - We thank the reviewer for this insightful comment. In our developed theory and ablation experiments **(Figure 5, Appendix G1)**, we demonstrate that increasing the projection dimension leads to a higher margin, which consequently improves performance. However, the performance stabilizes when the projection dimension reaches a specific value. Therefore, we can select a projection dimension that balances performance and computational overhead.
>
>   We agree that it would be ideal to theoretically infer the optimal projection dimension. However, this falls under the category of model selection, which remains a challenging problem in the field and, to the best of our knowledge, *still lacks* an efficient solution. This issue is analogous to determining the trade-off parameters for balancing multiple loss functions during the training of deep learning models, where the most practical approach is to fine-tune these parameters using a validation set.
>
> - Empirically, in this work, we use grid-search to choose the proper values of the dimension of projected space. In the table below, we provide experimental results to demonstrate the behavior of our proposed method (on Split-CIFAR-100), which can well offer insight for the community.
>
>   **[Table A]**
>   | Dimension | 5,000 | 10,000 | 15,000 | 20,000 | 25,000 |
>   | --- | --- | --- |---|---|---|
>   | Accuracy    |94.4 | 95.45| 96.01| 96.51| 96.52|
>   | Inference time (classification head only) ($\times 10^{-3}$  s) | 1.65| 3.31 | 4.25 | 4.88 | 5.27|
>   | #Params (classification head only, $\times 10^5$)| 5 | 10 | 15 | 20 | 25 |
>   | Memory (classification head only, MB)| 1.91|3.81 | 5.72| 7.63| 9.53|
>
> - Moreover, we would like to point out that our variant, without random projection and with  $K = 13$ views, significantly outperforms all state-of-the-art baselines. While it still requires determining the number of views $K$, we believe that selecting $K$ is considerably easier than setting the projection dimension.
>
> **3. The proposed self-improvement process for prompt selection could increase the model’s inference complexity, but this potential trade-off is not fully analyzed, especially in scenarios with many tasks.**
>
> - We provided this analysis in **Appendix G.3.3**. We would like to highlight that this strategy **does not** depend on a number of tasks, as it only involves iteratively choosing a more proper prompt during inference.
>
> **5. More experiment regarding hyper-parameters like number of views, projection dimensions, and threshold settings for the voting mechanism should be conducted.**
>
> - We provided these results regarding *the number of views, projection dimensions* in **Appendix G1**. In terms of *the threshold for the voting mechanism*, we would like to provide more detailed results as follows:
>
>   | Threshold | 0.2 | 0.5 | 0.6 | 0.7 | 0.9 |
>   | --- | --- | --- | ---|---|---|
>   | Split-CIFAR-100   | 94.65 | 95.45  | 95.03 | 94.21 | 94.03|
>   | Split-Imagenet-R   | 78.98 | 79.62  | 79.22  | 78.52|# |
>
> The results show that if the threshold is set too high (confident), the number of selected experts and the diversity of the corresponding answers will be limited, which in turn will restrict the model's performance.
>
> **6. How much GPUs resource does the proposed method need, will the GPU memory usage increase rapidly by increasing projection dimension.**
>
> For the experiments with the number of atomic view $K=15$ and the dimension of projected space $d'=10K$, we use *2 GPUs V100* as reported in **Appendix G.3**.
>
> Yes, GPU memory usage increases when increasing the projection dimension. Please refer to **Table A** (above) for the change in memory usage when the dimension of projected space varies.

---

> ### Author Response · Authors · 2024-11-22
> **Response to Reviewer gjF6 (2)**
>
> **1. There is limited discussion on the theoretical upper bound of generalization error.**
>
> Thanks for this solid suggestion. Before, we refered to some theoretical works such as  [1,2,3] for providing the link between the margin and the generalization ability. Our main theoretical contribution is to prove that the margin in high-dimensional projection space can increase at the rate of $O(\sqrt{d'})$ compared to the one in the original space. Here we would like to recap the pathway of our theory development with the link to the relevant theoretical upper bound of generalization ability.
>
> First, we define the instance margin. Given a classifier $g$, the   margin of an instance $s =(z,y)$ can be defined as
> $\gamma(s, g, \mathbb{R}^d) = \sup\\{\nu: \\|z - z' \\| \le \nu \Rightarrow g(z') = y, \forall z' \in  \mathbb{R}^d\\}$.
>
>
> Second, we define the margin over a training set. Given a training set $S$, we define the margin of $g$ over the training set $S$ as $\gamma(S,g,\mathbb{R}^d) = \min_{s\in S} \gamma(s,g,\mathbb{R}^d)$. The training set $S$ over $\mathbb{R}^d$ induces the training set $S^u$ over $\mathbb{R}^{d'}$ through the transform matrix $U$. Similarly, we define the margin over $S^u$.
>
> To see the benefits of RP, we propose the concept of Bayes margin to measure the highest margin of the classifiers in a hypothesis family.
>
> *Definition.* Let $\mathcal{H}$ be the set of measurable functions that map from  $\mathbb{R}^d$ to $\mathcal{Y}$. The Bayes margin over the training set $S$  is defined as
>     $\gamma_{\text{in}}(S,  \mathcal{H}) = \sup_{g \in \mathcal{H}} \gamma(S, g, \mathbb{R}^d)$.
>
> This Bayes margin refers to the classifier with the highest margin. Our Theorem 4.2 aims to prove that the *highest margin* in the projection space increases by the rate $O(\sqrt{d'})$ compared to the one in the original space. We then refer to some theoretical works typically [1, 2, 3], that develop the theories to demonstrate that higher margins lead to better generalization ability, to conclude that learning a classifier on the projection space can benefit its generalization ability. **Inspired by your suggestion, we've just added *Appendix A.1* to present a theory about generalization error to enhance the paper comprehensibility. For your convenient reference, we also present the theory in the post below.**
>
> **References:**
>
> [1] Spectrally-normalized margin bounds for neural networks, NeurIPS 2017.
>
> [2] Robust large margin deep neural networks. IEEE Transactions on Signal Processing, 2017
>
> [3] Robustness and generalization. Machine learning, 2012.

---

> ### Author Response · Authors · 2024-11-23
> **Response to Reviewer gjF6 (3)**
>
> **1. There is limited discussion on the theoretical upper bound of generalization error - (next)**
>
> Consider a *learning problem* specified by a model (or hypothesis) class $\mathcal{H}$, an instance set $\mathcal{X}$, and a loss function
> $\mathcal{l}:\mathcal{H} \times \mathcal{X} \rightarrow \mathbb{R}_{\geq 0}$,
> where each input instance $\mathbf{x} \in \mathcal{X}$  has a corresponding output $y \in \mathcal{Y}$. However, without loss of generality, we omit the output $y$  for simplicity of presentation.
>
> Given a distribution $P$ defined on $\mathcal{X}$, the quality of a model $\mathbf{h} \in \mathcal{H}$ is measured by its *expected loss* $F(P, \mathbf{h}) = E_{\mathbf{x} \sim P}[\mathcal{l}(\mathbf{h},\mathbf{x})]$. The *empirical loss*  $F(\mathbf{S}, \mathbf{h}) =  \frac{1}{|\mathbf{S}|} \sum_{\mathbf{x} \in \mathbf{S}} \mathcal{l}(\mathbf{h},\mathbf{x})$ is defined from a finite set
> $\mathbf{S} = \{ \mathbf{x}_1, ..., \mathbf{x}_n \} \subseteq \mathcal{X}$ of size $n$.
>
> Let $\Gamma(\mathcal{X}) := \bigcup_{i=1}^{K } \mathcal{X}_i$ be a partition of $\mathcal{X}$ into $K$ disjoint nonempty subsets. Denote $\mathbf{S}_i = \mathbf{S} \cap \mathcal{X}_i$ and $n_i = \| \mathbf{S}_i \|$ as the number of samples falling into
> $\mathcal{X}_i$.
>
>   (meaning that $n = \sum_{j=1}^K n_j$.) Denote $\mathbf{T} = \\{ i \in [K ] : \mathbf{S} \cap \mathcal{X}_i \ne \emptyset \\}$
>
> The following result is a simple consequence from algorithmic robustness[4, 3].
>
> **Theorem A.2:**
>    Consider a model $\mathbf{h}$ and a dataset $\mathbf{S}$ which consists of $n$ i.i.d. samples from distribution $P$. Let $\mathcal{l}(\mathbf{h},\mathbf{x})$ be the loss of $\mathbf{h}$ at instance $\mathbf{x}$, and $C_h = \sup_{\mathbf{x} \in \mathcal{X}} \mathcal{l}(\mathbf{h},\mathbf{x}), \epsilon(\mathbf{S}) = \sup_{i \in [K]}  \sup_{\mathbf{x} \in \mathcal{X}_i, \mathbf{s} \in \mathbf{S}_i} | \mathcal{l}(\mathbf{h},\mathbf{x}) -   \mathcal{l}(\mathbf{h},\mathbf{s}) |$. For any  $\delta >0$, the following holds  with probability at least $1-\delta$:
>   \begin{equation}
>   F(P, \mathbf{h}) \leq  F(\mathbf{S}, \mathbf{h}) +  \epsilon(\mathbf{S})  + 3C_h \sqrt{\frac{| \mathbf{T}| \ln(2 K /\delta)}{n}} + \frac{ 2 C_h | \mathbf{T}| \ln(2 K /\delta)}{n}
>  \end{equation}
> This theorem implies that the expected loss of a model can be bounded by using robustness level $\epsilon(\mathbf{S})$ of the model around the training samples. A more robust model can suggest better generalization on unseen data.
>
> Next we connect the margin and generalization ability of a model. Let $\mathcal{N}(\mathcal{X}, \\|\cdot \\|, \lambda)$ be the $\lambda$-*covering number* of $\mathcal{X}$ according to some metric $\\|\cdot \\|$ and $\Gamma(\mathcal{X})$ be the corresponding covering of $\mathcal{X}$, where $\lambda$ is a positive constant. This means $K = \mathcal{N}(\mathcal{X}, \\|\cdot \\|, \lambda)$. When the input space $\mathcal{X} \subseteq \mathbb{R}^d$ has diameter at most $R$, a classical fact [5] says that $\mathcal{N}(\mathcal{X}, \\|\cdot \\|, \lambda) \leq \left(\frac{2R\sqrt{d}}{\lambda}\right)^d$.
>
> Furthermore, Example 1 in [3] shows that $\epsilon(\mathbf{S}) =0$ for any $\lambda \le 0.5\gamma_h$ and 0-1 loss $\mathcal{l}$, where $\gamma_h = \max\\{v: \\|\mathbf{x} - \mathbf{s}\\| \leq v \Rightarrow \mathbf{h}(\mathbf{x}) = \mathbf{h}(\mathbf{s}) \text{ for all } \mathbf{x} \in \mathcal{X}, \mathbf{s} \in \mathbf{S}\\}$ denotes the margin of $\mathbf{h}$ for the training set.
> Combining those observations with **Theorem A.2** will lead to the following Corollary:
>
> **Corollary:**
> Given the notations in **Theorem A.2** with 0-1 loss $\mathcal{l}$ and $\mathcal{X} \subseteq \mathbb{R}^d$ with diameter at most $R$. Denote $\gamma_h = \max\\{v: \\|\mathbf{x} - \mathbf{s}\\| \leq v \Rightarrow \mathbf{h}(\mathbf{x}) = \mathbf{h}(\mathbf{s}) \text{ for all } \mathbf{x} \in \mathcal{X}, \mathbf{s} \in \mathbf{S}\\}$ as the margin of $\mathbf{h}$, and $A = \ln(2 /\delta) + d \ln(4R\sqrt{d}/\gamma_h)$. If $\gamma_h >0$, then the following holds  with probability at least $1-\delta$:
> \begin{equation}
>   F(P, \mathbf{h}) \le  F(\mathbf{S}, \mathbf{h})  + 3 \sqrt{\frac{| \mathbf{T}| A}{n}} + \frac{ 2  | \mathbf{T}| A}{n}
>  \end{equation}
> This result suggests that a model with larger margin can lead to smaller bound for the expected loss, suggesting better generalization ability. Combining this with the RP's ability to increase margin, we can conclude that RP can facilitate better generalization ability of a model in high dimensional spaces.
>
> **References:**
>
> [4] Robustness implies generalization via data-dependent generalization bounds, ICML 2022.
>
> [5] Yihong Wu and Yingxiang Yang. Lecture 14: Packing, covering, and consequences on minimax risk, 2016.

---

> > ### Comment · Reviewer_gjF6 · 2024-11-26
> >
> > Thanks for the authors' rebuttal, I decide to keep my score.

---

> ### Author Response · Authors · 2024-11-26
> **Looking forward to the detailed feedback from the Reviewer**
>
> Dear Reviewer gjF6,
>
> Thanks for your response. We believe that we have addressed well all your concerns. In particular,
> - For the first question, we have provided the generalization bound demonstrating the effectiveness of using Random Projection.
> - For the second question, regarding the theoretical guidance on choosing the optimal projection dimension in RP. We thoroughly explained that this is still a challenging problem in the field, and we open this for future work (as this not our main contribution). Alternatively, we provide the detailed experimental results showing the trade-off in terms of accuracy gained and computational cost when using higher projection dimension, which we believe this would offer insight for future work.
> - Regarding the ablation study and other details, almost the aspect of our method you required to be had provided in our manuscript. There was still only the experimental results regarding the threshold for voting mechanism, but we have fully provided afterward.
>
> Could you please let us know whether we've addressed your concerns and what we should do to further address them? Your detailed feedback will help enhance the quality of our work.
>
> *In addition, we would like to highlight that our method has theoretical support, the proposed approach is novel, presented meticulously and rigorously. Furthermore, our proposed approach achieves state-of-the-art performance and addresses the limitations of previous methods (e.g., RanPAC). We have also conducted comprehensive ablation studies to thoroughly investigate and showcase the behaviors of our approach. For these reasons, we believe our paper meets the criteria of a good paper, and we hope that our work will be more well acknowledged.*

---

> ### Author Response · Authors · 2024-11-28
> **Your further clarification is very important to us**
>
> Dear Reviewer gjF6,
>
> Thank you for your thoughtful comments and reviews, which have undoubtedly helped us improve our paper by addressing them. While we believe we have adequately addressed your concerns, we notice that you have expressed high confidence in your assessment of our work. Therefore, we would greatly appreciate it if you could clarify any specific points that you feel we have not sufficiently addressed, leading to your decision to maintain the current score (5). Your feedback would be invaluable in helping us further refine and enhance our paper.

---

> > ### Author Response · Authors · 2024-12-01
> > **Looking forward to your detailed response**
> >
> > Dear Reviewer gjF6,
> >
> > Thank you for your time reviewing our paper. As your feedback is critical to the paper decision, we kindly ask if you could provide more specific details on which points in our rebuttal you feel were not addressed adequately.
> >
> > Your feedback would greatly help us improve our work.
> >
> > Thank you again for your time and efforts.

---

### Official Review · Reviewer_WERc · 2024-11-03

**Soundness:** 3
**Presentation:** 3
**Contribution:** 3
**Rating:** 6
**Confidence:** 4

**Summary:**

This paper presents BoostCL, a method for pretrained-based continual learning that leverages multiple views and random projection (RP) to enhance performance. The authors theoretically analyze the benefits of RP in higher-dimensional spaces and demonstrate its effectiveness through experimental results.

**Strengths:**

1.	The paper provides a novel theoretical analysis of the benefits of random projection in higher-dimensional spaces for continual learning.
2.	The authors conduct a comprehensive set of experiments to evaluate the performance of their proposed method, BoostCL, across various tasks and datasets.
3.	The paper demonstrates the practicality of the proposed method by applying it to real-world continual learning scenarios.

**Weaknesses:**

1. The paper mentions the motivation behind using random projection, but it lacks a comprehensive discussion on why this approach is superior to other potential methods for feature transformation.
2. While the paper proposes a novel method for continual learning, it would be helpful to see a more in-depth comparison with other recent methods in the literature. Specifically, how does BoostCL differ from and improve upon existing approaches?
3. The technical details of the proposed method are somewhat challenging to follow. The paper could benefit from a more clear and concise explanation of the algorithm and its components.
4. Discussing the scalability of the proposed method to larger datasets or more complex models would be relevant.
5. The author needs to further analyze the consumption of the proposed method compared to other methods in terms of computational resources and training time.
6. The paper could benefit from a more organized and focused presentation of the material. The introduction could be more clearly tailored to motivate the problem and the proposed solution. Additionally, the paper includes several appendices that contain detailed proofs and additional experimental results. While these appendices provide valuable information, they could be more effectively integrated into the main text to enhance the clarity and readability of the paper.

**Questions:**

See weakness

---

> ### Author Response · Authors · 2024-11-22
> **Response to Reviewer WERc**
>
> We thank the reviewer for the constructive comments. We would like to address the main questions and concerns as follows:
>
> **1. The paper mentions the motivation behind using random projection, but it lacks a comprehensive discussion on why this approach is superior to other potential methods for feature transformation.**
>
> In the table below, we provide the experimental results (on Split-CIFAR-100) when applying various feature tranformation methods for learning classification head, including Random Projection, Log Transform, Square Transform, Reciprocal Transform, Yeo Johnson Transform.
> | Method | Random Projection (d'=10K) | W/o Random Projection|Random Projection (d'=d=768)|Log Transform (d'=768) | Square Transform (d'=768)|Reciprocal Transform (d'=768)|Yeo Johnson Transform (d'=768)|
> |---|---|---|---|---|---|---|---|
> |RanPAC|92.2 | 90.55 |90.56 | 1.01 |77.51|1.16 | 85.93
> |BoostCL | 96.55 | 94.67 | 94.67 | 1.02 | 78.23 | 1.05 | 87.02
>
> In particular, we can see that:
>
> + Compared to other feature transformation methods, RP is the most suitable one, which can help conveniently increase the dimension of features to a specific value.
>
> + In this work, as well as RanPAC, RP when projecting features into a significant higher-dimensional space (d'>>d), the margin of the classifier will increase accordingly, thereby enhancing model performance.
>
> + Besides, when apply RP without increasing the feature dimension, RP is the only method that does not harm the model performance as the rest.
>
> **2. While the paper proposes a novel method for continual learning, it would be helpful to see a more in-depth comparison with other recent methods in the literature.**
>
> - The main contributions of our work is our novel **ensemble classifier**. To the best of our knowledge, this is the **first ensemble-based method applied only to classification head**. Different from other existing ensemble methods for CL [1, 2, 3], our BoostCL is more elegant and efficient. In particular:
>     - Compared to the existing related methods, our method does not depend on a special design of the backbone, thereby we can flexibly apply it to any backbone to enhance model performance.
>     - Besides, our method does not need to forward many times through the model backbone to get response from each expert w.r.t each sample. Instead, BoostCL just works on the classification head, meaning that we can save the cost during inference.
>
> - The second technique is the **self-improvement process**. This is also the **first time considered** in the literature. Although it is simple, through iteratively predicting promptID, we can get the correct promptID with a higher probability and thus, enhance model performance compared to the existing baselines.
>
> - The third one is **integrating First-session adaptation strategy with the prompt-based backbone**, which is a combination of the strengths of existing approaches like RanPAC and HiDE, thereby helping improve model performance.
>
> [1] Divide and not forget: Ensemble of selectively trained experts in Continual Learning, ICLR2024.
>
> [2] Expandable Subspace Ensemble for Pre-Trained Model-Based Class-Incremental Learning, CVPR2024.
>
> [3] Boosting Continual Learning of Vision-Language Models via Mixture-of-Experts Adapters, CVPR2024.
>
> **4. Discussing the scalability of the proposed method to larger datasets or more complex models would be relevant.**
>
> - Regarding *datasets*, we have conducted experiments on a wide range of benchmark datasets in the setting of CL. The results consistently demonstrate the significant effectiveness and superiority of our method compared to the latest SOTA baselines.
>
> - Regarding *more complex models*, the superiority of our method compared to the baselines does not depend on the kind of architecture.
>     - Evidently, for the effectiveness of *our ensemble classifier*, we conducted experiments **(Table 5, Appendix G1)**, showing that our method can improve the performance of the existing baselines with a variety of strong levels and behaviors of the backbones.
>     - Besides, we would like to provide the additional results in the table below, showing that our method still consistently outperforms baselines:
>
> Using pre-trained ViT-B/16 + LoRA instead of prompt(prefix-tuning, as in the main paper):
> | | Split-CIFAR-100 | Split-Imagenet-R |
> | ---|---- | ---- |
> | HiDE | 92.92 | 75.58|
> | BoostCL | **96.62** | **80.54** |
>
> Using pretrained SWIN-L + prefix-tunning (instead of ViT-B/16, as in the main paper):
> | | Split-CIFAR-100 | Split-Imagenet-R |
> | --- | ---- | --- |
> | HiDE| 93.36| 77.05|
> | BoostCL| **96.85** | **80.89** |
>
> **5. The author needs to further analyze the consumption of the proposed method compared to other methods in terms of computational resources and training time.**
>
> - Regarding the **computational resources and training time**, we did thoroughly provided the experimental results, comparing ours with the two strongest baselines (HiDE and RanPAC) in **Appendix G3.**

---

> ### Author Response · Authors · 2024-11-22
> **Response to Reviewer WERc (2)**
>
> **3 and 6. Regarding the presentation of the paper**
>
> We thank the reviewer for your comments. We have revised our paper to enhance the readability.
>
> Regarding the experimental results, due to the length limit of the main text, we have included only the main and most important results. The remaining experiments, which relate to the thorough examination of other aspects of the method, are presented in the Appendix.

---

> > ### Author Response · Authors · 2024-12-02
> > **Your feedback is valuable to us**
> >
> > Dear Reviewer WERc,
> >
> > As the author-reviewer discussion period is ending shortly, we would greatly appreciate your feedback on whether our responses have addressed your questions and concerns. Thank you for your time and effort in reviewing our paper.

---

### Official Review · Reviewer_HTXd · 2024-11-03

**Soundness:** 3
**Presentation:** 3
**Contribution:** 2
**Rating:** 6
**Confidence:** 3

**Summary:**

This work considers the random projection (RP) strategy for pre-trained models in CL. Motivated by the benefits of RP in high-dimensional space, a multi-view strategy for an efficient classifier is further proposed, in which the principle of Adaboost is adapted to
overcome inherent obstacles and applied for the first time in CL. In addition, a self-improvement process technique, although simple, also shows significant effectiveness in selecting proper taskspecific prompts. The experimental results demonstrate a positive impact of the proposed method in improving model quality while only applying to linear classifiers.

**Strengths:**

(1)The proposed method BoostCL performs better than existing CL baselines, including Hide-prompt and ranpac.
(2)This work addressed the challenge when applied AdaBoost directly to CL.
(3)A self-improvement process, a simple but effective strategy is designed to help select prompts more accurately when inference.

**Weaknesses:**

As shown in Table 1, the proposed method takes no advantage in anti-forgetting, FFM metric.

**Questions:**

Does some ablation study be conducted on the proposed modules, Prompt selection process and Self-improvement process?

---

> ### Author Response · Authors · 2024-11-22
> **Response to Reviewer HTXd**
>
> We thank the reviewer for taking time to review and appreciating our paper. We would like to respond to the main questions and concerns as follows:
>
> **1. "As shown in Table 1, the proposed method takes no advantage in anti-forgetting, FFM metric."**
>
> We respectfully disagree that this is a significant weakness.
> - Regarding FFM, we would like to highlight that compared to baselines, our method achieves the best FMM scores on Split CIFAR-100 by a gap of 2% compared to the runner-up. For FFM on Split ImageNet-R, Split CUB-200, and 5-Datasets, BoostCL is the runner-up method, yet by a very small gap compared to the best one (0.12-0.88%).
> - Although there are cases in which ours does not reach the best FMM, we consistently obtain the highest score of FAA, by the impressive gap of 2-4% compared to the strongest baselines.
>
>
> **2. "Does some ablation study be conducted on the proposed modules, Prompt selection process and Self-improvement process?"**
>
> In our paper, we did provide comprehensive experimental results demonstrating the effectiveness of each component in our method. Specifically:
>
> - For the effectiveness of the **prompt selection process/self-improvement process**, please refer to the text from **Lines 515 - line 532**. Particularly, the results of "1 step" correspond to the conventional testing process, where the model first predicts promptID and then uses it to get a final prediction. Besides, the results of more than 1 step demonstrate the behavior of models with iterative steps of self-improvement.
>
> - We also provided the results corresponding to the "Strategy of **task-adaptive backbone**"; please refer to **Lines 504-514**.
>
> - Especially, regarding our main module **"Ensemble classification head"**:
>     - **In Table 1**, we provided detailed results in various cases, demonstrating the effectiveness and essence of our classification head in enhancing model performance, even when using fewer parameters than the baselines.
>     - Besides, other behavior regarding *computational time/ space* was carefully provided in **Table 2** and **Appendix G2, G3**),
>     - Moreover, the effectiveness of each component in this module *(number of atomic views and their dimension, our voting strategy, and applicability of our ensemble classifier)* was comprehensively presented in **Appendix G1**.

---

> > ### Comment · Reviewer_HTXd · 2024-11-25
> > **Thank you for your response**
> >
> > Thank you for your detailed response. While the updates and clarifications have improved the manuscript, I will maintain my current rating due to the inherent limitations of the FMM metric.

---

> ### Author Response · Authors · 2024-11-28
> **Addressing the final concern related to FFM**
>
> Dear Reviewer HTXd,
>
> Thank you for your response. We are glad that you feel we have adequately addressed all of your concerns, except for the one regarding FFM.
>
> For your concern related to FFM, we would like to provide a deeper analysis about this metric and explain why our method obtains significant higher FAA but has a bit higher FFM compared to few baselines (HiDEPrompt and CPP) in some cases.
>
> - First of all, in the Appendix F.3, we presented the definition FAA and FFM. In particular, we considered the accuracy on the $i$-th task after the model has been trained up to the $t$-th task, denoted as $A_{i,t}$. The average accuracy of all tasks observed up to the $t$-th task is calculated as $AA_{t} = \frac{1}{t}\sum_{i=1}^{t}A_{i,t}$. Upon the completion of all $T$ tasks, we report the Final Average Accuracy as $\text{FAA} = AA_{T}$. Additionally, we calculate the Final Forgetting Measure, defined as $\text{FFM} = \frac{1}{T - 1}\sum_{i=1}^{T - 1}\max_{t \in \{1, \ldots, T - 1\}}{(A_{i,t} - A_{i,T})}$.
>
>
>   ---> In this definition, FFM is the difference between the model performance at its best and worst (last) times
>
> - Based on these definitions, we would like to provide the experimental results in the image below, compare our method and the one of strongest baselines - HiDE on the Split ImageNet-R dataset where our FFM is 3.05 while the FFM of HiDE is 2.17.
>   - The subfigures in the first line shows the accuracy of  the models w.r.t data of tasks 3, 5, 7 over time. Specifically, we measure the performance over each of these tasks over the time. It can be observed that our BoostCL always achieves higher performance over the time from the first time and the last time learns each task (i.e. last model). Additionally, our line is always over the one of HiDE even the one at the last time, showing that our approach is better than HiDE when using the last or up-to-date model to predict previous tasks. In terms of FFM number, we are lower than HiDE due to our higher peak performances that lead to higher FFM gap. [The dashed lines are for illustrating the steep descent from the best to the worst results]
>
>   [Image: https://ibb.co/frMN5HD ]
>
>     - In addition, we argue that in Continual Learning, the FAA which reflects the performance when using the last model to predict previous tasks is much more important than FFM that only reflects the gap between the peak and last ones. The reason is that eventually, we need to use the last or up-to-date model to evaluate on current and previous tasks. As showed in the subfigures in the first line (the performance of our up-to-date models when predicting on old tasks that always outperforms HiDE) and second line (our FAA is always significantly superior HiDE by 2-4%), this demonstrates the advantages of our BoostCL when evaluating on all tasks so far using the up-to-date model.
>  - We agree that it is ideal if a CL model can obtain the highest peak performance and slowly reduce its performance over time on a task. That is the reason why we need FFM. Regarding FFM, ours and baselines are comparable, which we are lower than some cases and higher in some cases (the gap is from 0.2-0.88%). However, our average FFM is better than the most outstanding baseline HiDE (ours is 1.658 and HiDE is 1.905).
>
> Finally, as our analysis, our BoostCL is significantly better than baselines when using the last or up-to-date model to predict all task so far, which is the main objective of CL reflecting through the metric FFA. Regarding the FFM, we are comparable to the baselines and gain better overall average performance. Your comment suggests us to further enhance FFM, which we would like to leave for a future work. We really appreciate if the reviewer considers raising your rating given the fact that ours has theory support, archives the-state-of-the-art performance together with comprehensive additional and ablation experiments we conducted.

---

> > ### Author Response · Authors · 2024-12-02
> > **Addressing the final concern related to FFM**
> >
> > Dear Reviewer HTXd,
> >
> > As the author-reviewer discussion period is coming to an end shortly, we would greatly appreciate your feedback on whether our response has addressed your concerns regarding FFM. Thank you for your time and effort in reviewing our paper.

---

### Author Response · Authors · 2024-11-24
**Looking forward to the responses from Reviewers**

Dear Reviewers,

We would like to thank you again for spending your time evaluating our paper.

As the discussion period is expected to conclude shortly, we look forward to hearing your feedback about whether we have addressed your concerns in the rebuttals.

Best regards,

Authors

---

### Meta-Review · Area_Chair_gXmi · 2024-12-23

**Metareview:**

The paper introduces BoostCL, a method that outperforms existing continual learning (CL) baselines like Hide-prompt and ranpac by addressing the direct application challenges of AdaBoost to CL. On the positive side, the work provides a novel theoretical analysis on the benefits of random projection (RP) in high-dimensional spaces, which improves feature separability and supports continual learning. However, reviewers also raised several major concerns, including the weakness in anti-forgetting performance, technical and writing clarity, model scalability, theoretical analysis (e.g., upper bounds of generalization error), and lack of comparisons with the baseline methods.

**Additional Comments On Reviewer Discussion:**

The authors' responses addressed most of the concerns of the reviewers, while reviewers chose to maintain their initial scores. It should be noted that the authors were able to address most of the concerns of the reviewer gjF6, but did not receive detailed responses from the reviewer. The discussion and revised paper convinced the AC.

---

### Decision · Program_Chairs · 2025-01-22

Accept (Poster)